# High neural activity accelerates the decline of cognitive plasticity with age in *Caenorhabditis elegans*

**Qiaochu Li[1], Daniel-Cosmin Marcu[1], Ottavia Palazzo[1], Frances Turner[2], Declan King[1,3], Tara L Spires-Jones[1,3], Melanie I Stefan[1,4], Karl Emanuel Busch[1]\***

[1]Centre for Discovery Brain Sciences, Edinburgh Medical School: Biomedical Sciences, The University of Edinburgh, Edinburgh, United Kingdom; [2]Edinburgh Genomics (Genome Science), Ashworth Laboratories, The University of Edinburgh, Edinburgh, United Kingdom; [3]United Kingdom Dementia Research Institute, The University of Edinburgh, Edinburgh, United Kingdom; [4]ZJU-UoE Institute, Zhejiang University, Haining, China

**Abstract** The ability to learn progressively declines with age. Neural hyperactivity has been implicated in impairing cognitive plasticity with age, but the molecular mechanisms remain elusive. Here, we show that chronic excitation of the *Caenorhabditis elegans* $O_2$-sensing neurons during ageing causes a rapid decline of experience-dependent plasticity in response to environmental $O_2$ concentration, whereas sustaining lower activity of $O_2$-sensing neurons retains plasticity with age. We demonstrate that neural activity alters the ageing trajectory in the transcriptome of $O_2$-sensing neurons, and our data suggest that high-activity neurons redirect resources from maintaining plasticity to sustaining continuous firing. Sustaining plasticity with age requires the $K^+$-dependent $Na^+/Ca^{2+}$ (NCKX) exchanger, whereas the decline of plasticity with age in high-activity neurons acts through calmodulin and the scaffold protein Kidins220. Our findings demonstrate directly that the activity of neurons alters neuronal homeostasis to govern the age-related decline of neural plasticity and throw light on the mechanisms involved.

**\*For correspondence:**
emanuel.busch@ed.ac.uk

## Introduction

One of the most feared aspects of ageing is that as we get older our brain function deteriorates. Neural plasticity, which is the ability of the nervous system to adapt to changing environments or to encode new memories, declines continuously and progressively from early adulthood (*Arey and Murphy, 2017*; *Park and Bischof, 2013*). The environmental and intrinsic factors driving this decline of neural plasticity are not well understood (*Arey and Murphy, 2017*; *Bénard and Doitsidou, 2017*; *Bishop et al., 2010*; *Stein and Murphy, 2012*). Sensory input and neural excitation are important in regulating the ageing process and have been shown to control *C. elegans* lifespan (*Alcedo et al., 2013*; *Zullo et al., 2019*), but how they drive the ageing of cognitive function and plasticity remains obscure.

A growing body of evidence indicates that neural hyperactivity is an early-stage functional hallmark of neurodegeneration, particularly in Alzheimer's disease (*Busche and Konnerth, 2016*; *Filippini et al., 2009*; *Hämäläinen et al., 2007*; *Koelewijn et al., 2019*; *Leal et al., 2017*; *Palop et al., 2007*; *Putcha et al., 2011*; *Stargardt et al., 2015*). Other studies too suggest that neural hyperactivity impairs memory formation. For example, subregions of the hippocampus with elevated neural activity are unable to encode new information in ageing rats (*Wilson et al., 2005*), and drug treatment that lowers hippocampal activity improves memory formation in humans (*Bakker et al., 2012*). Also, epilepsy, which is characterized by neural overactivation, is frequently

associated with cognitive impairment, especially with deficits in learning and memory, and increases the risk of developing dementia with age (*Dodrill, 2002*; *Hermann et al., 2008*; *Miller et al., 2016*; *Sen et al., 2018*). A direct role of sensory input and neuronal activity in age-related cognitive decline has not yet been studied, however. Dysregulation of neuronal $Ca^{2+}$ signaling has been implicated in activity-linked cognitive decline (*Foster, 2007*; *Lerdkrai et al., 2018*; *Pchitskaya et al., 2018*; *Toescu and Verkhratsky, 2007a*), but the underlying genetic and molecular mechanisms are poorly understood. Elucidating how neuronal activity regulates cognitive decline at the molecular level may pave the way for a strategy to slow cognitive decline with age.

To study the mechanisms that govern the decline of cognitive plasticity with age, we are using the robust and reproducible behavioral responses to $O_2$ in *C. elegans,* a genetically and molecularly tractable nematode worm with an average lifespan of three weeks. These animals strongly avoid environments in which the atmosphere is either rich in $O_2$ (21%, as in the air) or poor in $O_2$ (<5%) and they move toward environments with $O_2$ levels of 5–10%, which are optimal for adequate cellular respiration but cause little oxidative stress (*Busch et al., 2012*; *Gray et al., 2004*; *Lee and Atkinson, 1976*; *Persson et al., 2009*). As the concentration of $O_2$ increases, the animals gradually switch their locomotory behavior from low-speed 'dwelling' to high-speed 'persistent roaming'. This $O_2$-evoked switch is sustained for as long as the animals are exposed to the higher concentration of $O_2$ (*Busch et al., 2012*). Four $O_2$-sensing neurons – the URX pair, AQR and PQR – continuously respond to high concentrations of $O_2$ to generate this persistent roaming behavior (*Busch et al., 2012*). Tonic activity of these neurons is necessary and sufficient to set the behavioral state according to the ambient $O_2$ concentration for many minutes and even hours. The activity state and cytoplasmic $Ca^{2+}$ concentration of these neurons are chronically elevated by long-term exposure to a high concentration (21%) of $O_2$ (*Busch et al., 2012*). The functional properties of these $O_2$-sensing neurons thus make them an ideal system for manipulating long-term neuronal excitation in vivo.

Here, we have established an assay in *C. elegans* to elucidate how neuronal activity affects the decline of neural plasticity with age. Animals show $O_2$ experience-dependent behavioral plasticity, where an overnight shift of the $O_2$ concentration in the culture environment reprograms the worms' $O_2$-evoked speed responses. We show that long-term activation of $O_2$-sensing neurons accelerates the decline of plasticity with age at both the neuronal and behavioral level. By gene expression profiling of $O_2$-sensing neurons in ageing animals, we show that neuronal activity alters age-related changes in the transcriptome of $O_2$-sensing neurons, suggesting a re-distribution of neuronal resources according to neuronal activity state during ageing. In particular, the differential expression of neuronal genes that modulate $Ca^{2+}$ homeostasis plays a central role in mediating activity-dependent decline. Low activity neurons require the $K^+$-dependent $Na^+$/$Ca^{2+}$ (NCKX) exchanger to remove intracellular $Ca^{2+}$ in order to sustain plasticity with age, whereas the decline of plasticity associated with high neuronal activity acts through calmodulin and the scaffold protein Kidins220 (also known as ARMS).

## Results

### Previous oxygen experience determines whether the plasticity of $O_2$ responses is maintained or lost with age in *C. elegans*

Long-term exposure to different oxygen environments creates a memory that configures *C. elegans'* aversion to high $CO_2$ (*Fenk and de Bono, 2017*). We therefore hypothesized that *C. elegans* adapt their $O_2$ preference depending on previous oxygen experience. To test this, we cultured animals from birth in an atmosphere of either their preferred $O_2$ concentration (7%) or at 21% $O_2$, and assayed the speed of locomotion of young (1-day-old) adults when they were exposed to a series of stepwise decreasing concentrations from 21% to 7% $O_2$ and a final step from 7% to 21% $O_2$ (*Figure 1—figure supplement 1A,B*). The reference laboratory strain N2 has acquired a hyperactive version of the neuropeptide receptor NPR-1, which attenuates avoidance of 21% $O_2$ in the presence of bacterial food (*McGrath et al., 2009*; *Persson et al., 2009*). We therefore tested animals bearing the *ad609* loss-of-function mutation in *npr-1*, thus restoring their robust avoidance of high $O_2$ (*Busch et al., 2012*). Strikingly, the concentration of $O_2$ at which the worms were initially cultured determined their responses to the stepwise changes in $O_2$ concentration: animals initially cultured in 21% $O_2$ gradually decreased their speed of locomotion as the $O_2$ concentration decreased from

21% to 7%, as observed in previous studies (*Busch et al., 2012*); those initially cultured in 7% $O_2$, however, were highly motile at all concentrations from 21% to 11% and only exhibited dwelling behavior when they reached their initial culture condition of 7% $O_2$ (*Figure 1—figure supplement 1B*).

We hypothesized that animals can 'update' their $O_2$ preference if they have recently been switched to a different oxygen environment, and tested this by transferring them from 21% to 7% $O_2$ or vice versa 12 hr before assaying their locomotory speed in response to stepwise changes in $O_2$ concentration (*Figure 1A*). Indeed, when 1-day-old adults cultured at 21% were shifted to 7% $O_2$ overnight, their locomotory response to intermediate $O_2$ strongly increased compared to animals cultured at high oxygen throughout, and resembled that of animals that had been cultured at 7% $O_2$ for their whole life (*Figure 1B,C*). Conversely, animals shifted from 7% $O_2$ to a 21% oxygen environment decreased their locomotory speed at intermediate $O_2$ relative to animals cultured at 7% oxygen throughout, and their response resembled that of animals cultured at 21% $O_2$ for their whole life (*Figure 1B,C*). This shows that the tuning of oxygen responses depends on the $O_2$ conditions animals were cultured in and is plastic according to the most recent $O_2$ concentration change.

To investigate how age might affect this oxygen-evoked behavior, we cultured animals at either 21% or 7% $O_2$ up to the adult age of 4, 7, or 10 days and tested them in the same assay as above. In both culture conditions, locomotory speed at 21% $O_2$ gradually declined with age, from 163 µm/s for 1-day-old to 97 µm/s for 10-day-old animals cultured in 21% $O_2$ and from 188 µm/s for 1-day-old to 114 µm/s for 10-day-old animals cultured in 7% $O_2$; these speeds still enabled animals to robustly avoid high oxygen environments, however (*Figure 1B,C,E,F,H,I* and *Figure 1—figure supplement 1C,E*). This suggests that the sensory ability itself remains largely intact until day 10 of adulthood but is accompanied by a progressive decline in locomotory ability, consistent with previous studies on the decline of *C. elegans* nervous system function with age, where locomotory speed typically reaches half-maximal capacity in 7–8 day-old adults (*Glenn et al., 2004*; *Hahm et al., 2015*; *Hsu et al., 2009*; *Podshivalova et al., 2017*). Response tuning by $O_2$ culture condition, that is, the higher locomotory speed at intermediate $O_2$ concentrations of worms cultured at 7% $O_2$ and lower speed of worms cultured at 21% $O_2$, is preserved in ageing animals (*Figure 1B,C* and *Figure 1—figure supplement 1C,E*). We then tested if experience-dependent plasticity of $O_2$ responses declines with age, by switching 3-, 6-, and 9-day-old animals to the opposite $O_2$ environment for 12 hr and recording $O_2$-evoked locomotion the next day (*Figure 1A*). Animals that were cultured at high ambient $O_2$ and switched to 7% started to lose the ability to update their $O_2$-evoked responses already at day 4 and by day 7 showed no difference to those of animals kept continuously at 21% (*Figure 1B,D*, *Figure 1—figure supplement 1D*). By contrast, animals cultured at 7% $O_2$ largely retained $O_2$ experience-dependent plasticity even up to day 10 of adulthood and changed their locomotory response to stepwise changes in $O_2$ concentration after switching to 21% $O_2$ for 12 hr (*Figure 1C,D*, *Figure 1—figure supplement 1F*). We calculated a plasticity index based on the sum of the speed differential across measurement points, normalized to the plasticity in young adults (see Materials and methods for details). The index documents the divergence in plasticity for worms exposed to a high or low oxygen environment (*Figure 1D*). The decline of cognitive plasticity with age is thus conditional on environmental state.

To test whether the plasticity loss in animals shifted from 21% to 7% $O_2$ and the plasticity retention in animals shifted from 7% to 21% $O_2$ may be a consequence of a bias in the assay paradigm, we compared the plasticity of 21% and 7% $O_2$ cultured animals after animals were shifted from 21% to 7% $O_2$, or from 7% to 21% $O_2$ (*Figure 1—figure supplement 1G,H*). Animals first cultured at 21% $O_2$ and then cultured for 1 day at 7% $O_2$ showed significantly less plasticity after the 7%→21% $O_2$ shift than animals continuously kept at 7% $O_2$ (*Figure 1—figure supplement 1G,I*), while animals first cultured at 7% $O_2$ and then cultured for 1 day at 21% $O_2$ show significantly higher plasticity after the 21%→7% $O_2$ shift than animals continuously kept at 21% $O_2$ (*Figure 1—figure supplement 1H, I*). This suggests that the decline rate of plasticity is dictated by the accumulated long-term environmental exposure to high or low $O_2$, rather than the direction of the $O_2$ shift. In addition, we observed that the plasticity of 7-day-old 21% $O_2$-raised animals cultured for 1 day at 7% $O_2$ before the plasticity experiment is significantly higher than that of animals kept at 21% $O_2$ throughout (*Figure 1—figure supplement 1G–I*). This suggests that the loss of plasticity in ageing 21% $O_2$-exposed animals is not irreversible but can be restored by temporary cultivation at low oxygen.

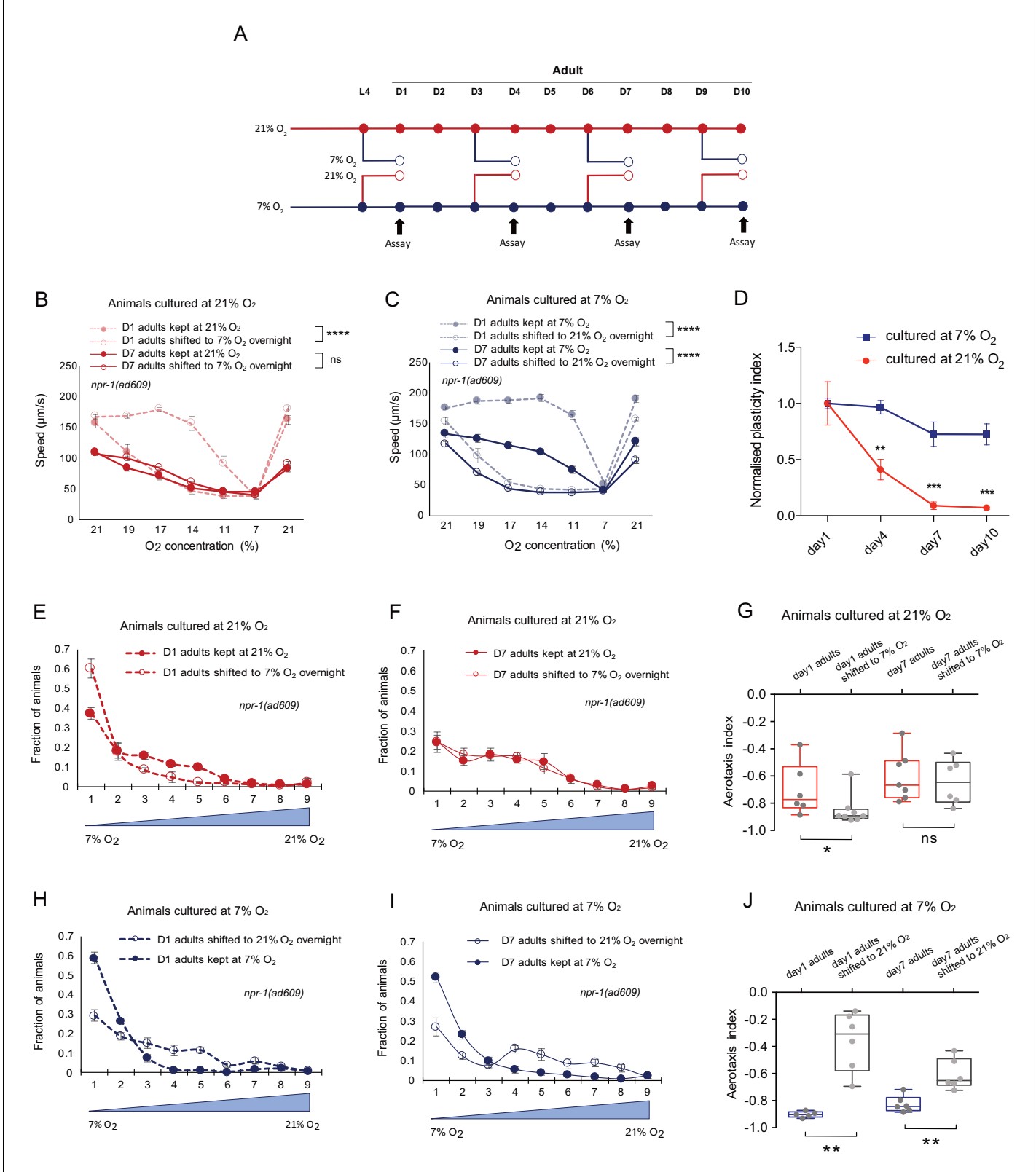

**Figure 1.** Previous oxygen experience determines the retention or loss of the plasticity of O$_2$ responses with age in *C. elegans*. (**A**) Assay scheme for testing O$_2$-evoked speed responses of day 1, day 4, day 7, and day 10 adults. (**B and C**) O$_2$-evoked speed responses of day 1 and day 7 adults cultured at 21% O$_2$ (**B**) and 7% O$_2$ (**C**), and speed responses after shifting to 7% O$_2$ (**B**) and 21% O$_2$(**C**) for overnight culturing at L4 stage and day 6 of adulthood respectively. Mean ± sem, n = 7 assays (70–140 animals) were performed for each condition, ****p<0.0001 indicates a significant effect of overnight O$_2$

*Figure 1 continued on next page*

*Figure 1 continued*

level shift on speed responses, ns, p>0.05, mixed model ANOVA. (D) Animals cultured at 7% $O_2$ showed slower decline rate of plasticity with age compared to animals cultured at 21% $O_2$. Mean ± sem, n = 3–7 assays (45–140 animals) for each condition, **p<0.01, ***p<0.001, asterisks indicate a significant difference between the normalized plasticity index of 21% and 7% $O_2$ cultured animals, unpaired t-test with Holm-Sidak correction for multiple comparisons. (E and F) Aerotaxis assay showing $O_2$ preference of day 1 (E) and day 7 adults (F) cultured at 21% $O_2$ and after shifting to 7% $O_2$ for overnight culturing. Animals were exposed to an $O_2$ gradient from 7% to 21% $O_2$ and the area was divided into nine grids for counting and calculation. Mean ± sem, n = 6–8 assays. (G) Aerotaxis index of day 1 adults cultured at 21% $O_2$ showed plasticity after shifting to 7% $O_2$, while day 7 adults cultured at 21% $O_2$ showed no plasticity after shifting to 7% $O_2$. Mean ± sem, *p<0.05, ns, p>0.05, Mann-Whitney U test. (H and I) Aerotaxis assay showing $O_2$ preference of day 1 (H) and day 7 adults (I) cultured at 7% $O_2$ and after shifting to 21% $O_2$ for overnight culturing. Mean ±sem, n = 5–6 assays. (J) Aerotaxis index of day 1 and day 7 adults cultured at 7% $O_2$ showed plasticity. Mean ± sem, **p<0.01, Mann-Whitney U test.

The online version of this article includes the following figure supplement(s) for figure 1:

**Figure supplement 1.** Behavioral assay program, $O_2$-evoked speed responses during ageing, and assay paradigm validation.

Previous experience also alters spatial oxygen preferences, and, like speed, this spatial preference can be updated by recent $O_2$ shifts. In a 7–21% $O_2$ gradient, animals previously cultured at 7% $O_2$ accumulated in a narrow range of $O_2$ near 7%, whereas those acclimated to 21% $O_2$ distributed over a broader range of concentrations between 7% and near 21% $O_2$, showing attenuated avoidance to high and intermediate $O_2$ concentrations (*Figure 1E,H*). These different preferences are maintained in 7-day-old adults (*Figure 1F,I*). Consistent with the differential decline of $O_2$-evoked locomotory responses, day 1 adults cultured at either 21% or 7% $O_2$ both show plasticity in their oxygen preference after switching culture conditions (*Figure 1E,H,G,J*), but only 7% $O_2$ cultured worms retained plasticity at day 7 of adulthood (*Figure 1F,I,G,J*).

Taken together, these results indicate that previous $O_2$ experience generates a memory that sculpts $O_2$-evoked behavioral responses. The ability to update this memory declines with age conditional on the previous $O_2$ environment: it is lost in ageing animals cultured at 21% $O_2$ but maintained in animals kept at 7% $O_2$.

## Plasticity of $O_2$-evoked $Ca^{2+}$ responses is lost in ageing neurons chronically stimulated with high $O_2$

To investigate how previous experience and age alter the neural encoding of $O_2$ levels, we recorded $Ca^{2+}$ responses in URX neurons in vivo by using the $Ca^{2+}$ sensor cameleon YC3.60 when 1- and 7-day-old adults were exposed to stepwise changing $O_2$ concentrations, with the same $O_2$ steps as in the assay testing the $O_2$-evoked locomotory speed. The URX oxygen-sensing neuron pair are tonic receptors that continuously signal ambient $O_2$ concentration, sufficient to mediate most $O_2$-evoked behaviors, including the regulation of locomotion, aerotaxis and aggregation (*Busch et al., 2012*; *Macosko et al., 2009*). Consistent with previous results (*Busch et al., 2012*), a 21% $O_2$ stimulus induced a tonically elevated $Ca^{2+}$ response in URX, whereas at 7% $O_2$ the intracellular $Ca^{2+}$ concentration dropped to baseline. Responses to $O_2$ were graded and declined with decreasing ambient $O_2$ concentration (*Figure 2A,B*). $O_2$ concentration in the culture environment profoundly affected the magnitude of $O_2$-evoked $Ca^{2+}$ responses: in animals grown at 7% $O_2$, URX displayed significantly higher [$Ca^{2+}$] responses to high and intermediate $O_2$ concentrations compared to 21% $O_2$ cultured animals (*Figure 2A*, red trace and *Figure 2B*, blue trace, *Figure 2—figure supplement 1C*).

To test whether the $Ca^{2+}$ responses of URX neurons to $O_2$ reflect the plasticity seen in $O_2$-dependent locomotory behavior, we transferred animals from 21% to 7% $O_2$ or vice versa 12 hr before the $Ca^{2+}$ imaging in one day old adults. Animals shifted from 21% to 7% $O_2$ showed a significantly increased URX responses to the $O_2$ stimuli compared to animals kept at 21% $O_2$ throughout (*Figure 2A*; *Figure 2—figure supplement 1A*; *Figure 1B*). Conversely, 1-day-old animals shifted from 7% to 21% $O_2$ showed significantly reduced responses to $O_2$ stimuli compared to animals kept at 7% $O_2$ (*Figure 2B*; *Figure 2—figure supplement 1A*; *Figure 1C*). There was no significant difference between the URX $Ca^{2+}$ responses in worms shifted from 21% to 7% $O_2$ (*Figure 2A*, blue line) and those of animals that have been grown at 7% $O_2$ only (*Figure 2B*, blue line; mixed-model ANOVA, p=0.2682). Therefore, $Ca^{2+}$ responses of URX neurons in young adults to $O_2$ are reprogrammed by switching the animals to a different $O_2$ environment.

We next tested how age affects URX activity and plasticity. In 7-day-old adults cultured at 7% $O_2$, $Ca^{2+}$ responses to high $O_2$ were significantly reduced in magnitude when compared to those of 1-

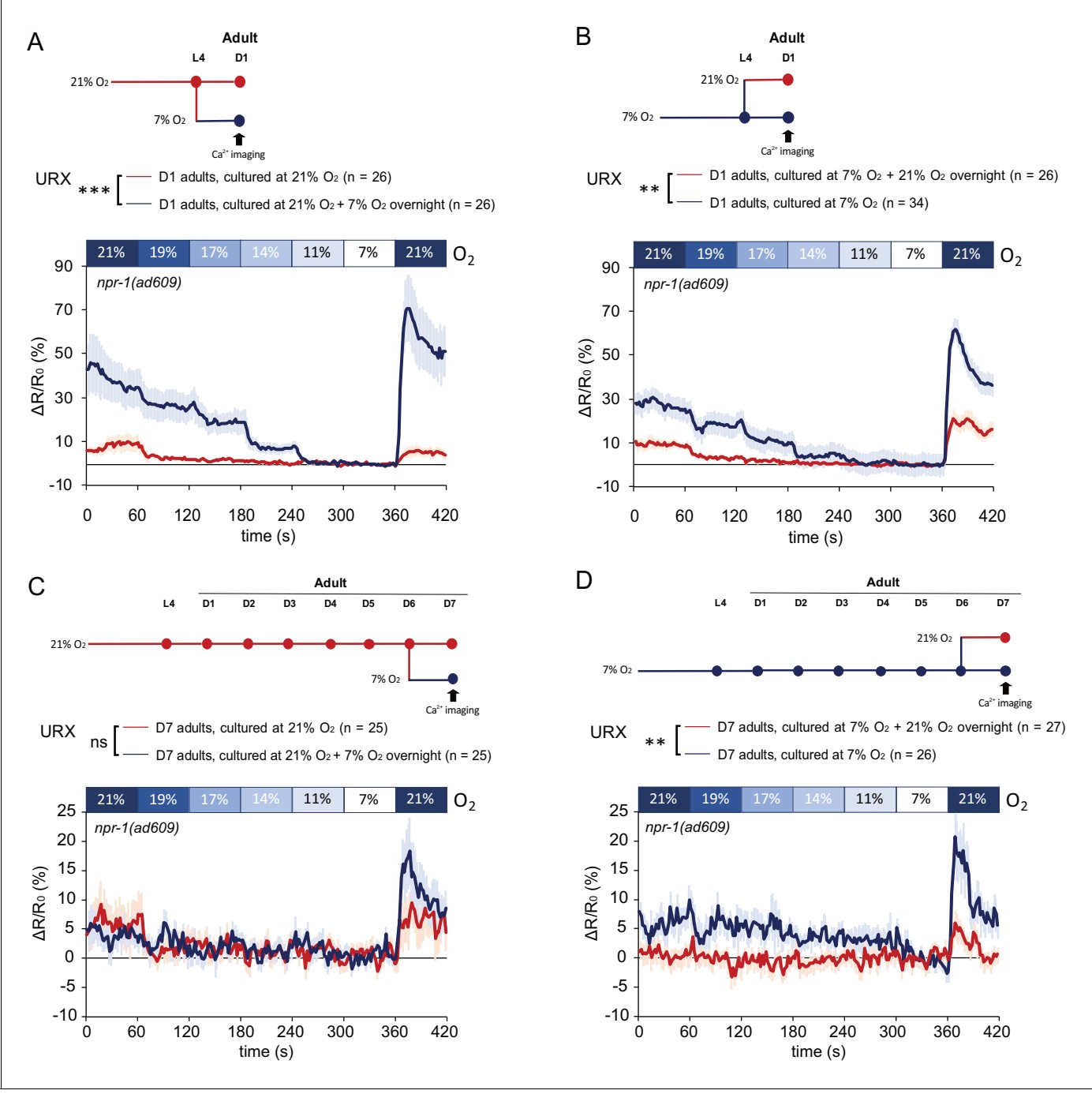

**Figure 2.** Plasticity of $O_2$-evoked $Ca^{2+}$ responses in URX of day7 adults is lost when neurons are chronically stimulated with 21% $O_2$, but is retained when neurons are persistently inactive at 7% $O_2$. (A and B) $Ca^{2+}$ responses of URX to $O_2$ of day one adults grown at 21% (A) and 7% $O_2$ (B), and $Ca^{2+}$ responses of animals shifted to the opposite $O_2$ condition the night before imaging. $O_2$ was applied in the following order the same as speed assay: 21%, 19%, 17%, 14%, 11%, 7%, and 21%. Each step lasts for 1 min. Mean ± sem, n = 26–34 animals per condition, for each 1 min step, 11–60 s period was used for statistical analysis, **p<0.01, ***p<0.001, asterisks indicate a significant effect of overnight $O_2$ level shift on $Ca^{2+}$ responses, mixed model ANOVA. (C and D) $Ca^{2+}$ responses of URX to $O_2$ of day 7 adults grown at 21% (C) and 7% $O_2$ (D), and $Ca^{2+}$ responses of animals shifted to the opposite $O_2$ condition the night before imaging. Mean ±sem, n = 25–27 animals per condition, for each 1 min step, 11–60 s period was used for statistical analysis, **p<0.01, ns, p>0.05, asterisks indicate a significant effect of overnight $O_2$ level shift on $Ca^{2+}$ responses, mixed model ANOVA. The online version of this article includes the following figure supplement(s) for figure 2:

**Figure supplement 1.** Scatter plots showing URX $Ca^{2+}$ responses of day 1 and day 7 adults cultured at 21% or 7% $O_2$.

day-old adults, whereas the responses of animals kept at 21% $O_2$ were maintained at similar levels to those of young adults (*Figure 2—figure supplement 1D*, *Figure 2C*, red trace and *Figure 2D*, blue trace). When the ageing 21% $O_2$ cultured animals were shifted to 7% for 12 hr, by contrast, the plasticity of the URX response was lost entirely (*Figure 2C*; *Figure 2—figure supplement 1B*). In contrast, 7-day adults cultured at 7% $O_2$ retain the URX plasticity observed in young adults where $Ca^{2+}$ responses to the series of oxygen stimuli are reduced after being shifted to 21% $O_2$ for 12 hr (*Figure 2D*; *Figure 2—figure supplement 1B*).

Taken together, these results show that the URX $Ca^{2+}$ responses are programmed by previous $O_2$ experience in the culture environment, and display strong experience-dependent plasticity in response to recently changed ambient $O_2$ concentration. Moreover, previous experience governs the age-dependent decline of both neuronal and behavioral plasticity: exposure to high ambient oxygen causes their decline and loss within 1 week, whereas at lower oxygen levels plasticity is well preserved in 7-day-old animals.

## The accelerated plasticity decline in ageing high $O_2$-cultured animals is caused by chronically high neuronal activity but not organismal oxidative stress

The loss of experience-dependent plasticity in ageing animals cultured in a high $O_2$ environment may result from either tonically high activity and elevated $[Ca^{2+}]$ in the $O_2$-sensing neurons (*Busch et al., 2012*) or increased oxidative stress in the organism (*Jagannathan et al., 2016*).

To test if chronically elevated neural activity is responsible for the accelerated decline of plasticity with age, we inhibited the activity of the oxygen-sensing neurons URX, AQR and PQR long-term while exposing them to the high oxygen condition. To do so, we expressed a chemogenetic tool, the *Drosophila* histamine-gated chloride channel HisCl1, in the $O_2$-sensing neurons. Addition of exogenous histamine to *C. elegans* culture plates enables the rapid, reversible and prolonged inhibition of neurons expressing this channel (*Pokala et al., 2014*). We cultured *pgcy-32::HisCl1* animals and control animals lacking the transgene in 21% oxygen from birth until 6 days old in the presence of 20 mM histamine, and *pgcy-32::HisCl1* animals grown in 21% $O_2$ in the absence of histamine. They were then either switched to 7% $O_2$ for 18 hr or retained at 21% $O_2$ (*Figure 3A*). Animals with inhibited $O_2$-sensing neurons showed strong plasticity after switching them to 7% $O_2$, whereas control animals grown without histamine or not expressing HisCl1 showed little plasticity (*Figure 3B*). Plasticity of the 7-day-old animals with inhibited neurons was nearly as great as that of 1-day-old animals and significantly greater than in either of the control conditions, whereas histamine treatment or HisCl1 expression itself did not prevent the decline of plasticity with age (*Figure 3C* and *Figure 3—figure supplement 2D*). In contrast, experience-dependent plasticity of 7-day-old adults cultured at 7% $O_2$, where the $O_2$-sensing neurons are not chronically active, was not altered by HisCl1-mediated silencing of the $O_2$-sensing neurons (*Figure 3—figure supplement 1A–C,E*).

The chemogenetic silencing of the $O_2$-sensing neurons shows that high neural activity is *necessary* to accelerate the decline of plasticity. To investigate if high neural activity is *sufficient* to accelerate cognitive decline, we performed chronic optogenetic stimulation of the URX $O_2$-sensing neurons in animals cultured at 7% $O_2$, using the blue light-gated cation channel Channelrhodopsin (ChR) (*Figure 3D–F*). We tested 4-day-old adults. At this age, experience-dependent plasticity is sustained in animals grown at 7% $O_2$, but declines in animals cultured at 21% $O_2$ (*Figure 1D*). Animals cultured at 7% $O_2$ with chronic URX excitation showed significantly reduced behavioral plasticity at day 4 of adulthood, compared to controls (ChR-expressing animals exposed to blue light without the administration of the rhodopsin cofactor all-*trans* retinal) (*Figure 3D,F*). In addition, chronic blue light illumination in the presence of all-*trans* retinal (ATR) did not significantly affect plasticity in control animals without ChR expression (*Figure 3E,F*). The chronic stimulation only alters the behavior of animals that are switched to 21% $O_2$ but not that of animals kept at 7% $O_2$ throughout (*Figure 3D*). This suggests that chronic optogenetic activation of URX does not alter $O_2$ responses per se, but specifically changes how the plasticity of the $O_2$ response declines with age.

To investigate the potential role of oxidative stress, we also performed the behavioral plasticity assay in ageing *C. elegans* cultured in the presence of the antioxidant N-acetyl cysteine (NAC), which increases resistance to oxidative stress (*Desjardins et al., 2017*). NAC treatment did not restore behavioral plasticity to 7-day-old animals cultured at high oxygen levels, nor did it have an effect on 1- and 7-day-old adults cultured in 7% $O_2$ that do show oxygen-evoked behavioral plasticity

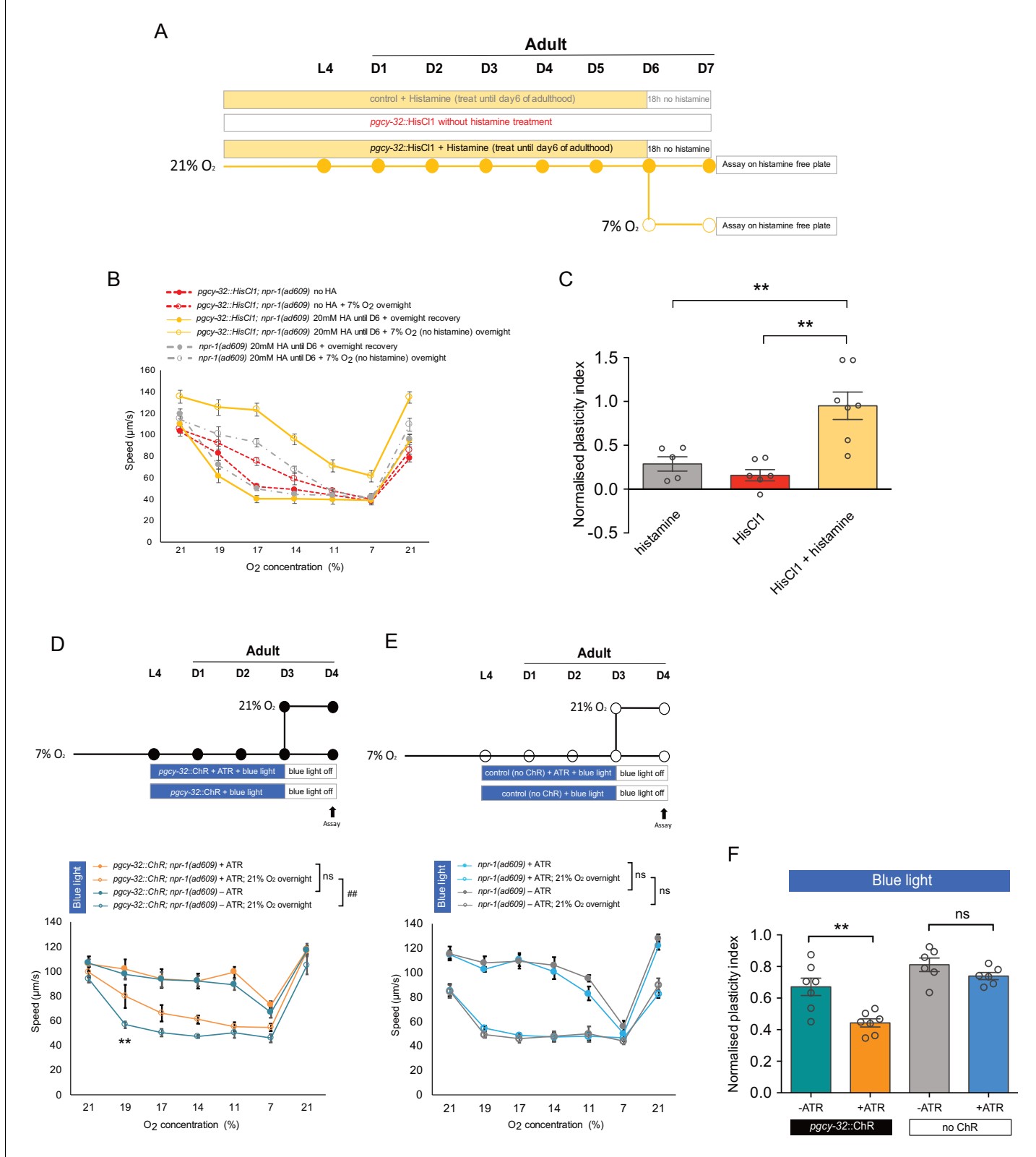

**Figure 3.** Long-term inhibition of neuronal activity restores, whereas long-term neuronal excitation accelerates the decline of age-related plasticity. (A) Assay scheme of long-term silencing of $O_2$-sensing neurons by histamine induced HisCl1 in $O_2$-sensing neurons. (B) $O_2$-evoked speed responses of *pgcy-32::HisCl1* animals with and without 20 mM histamine treatment, and control animals with 20 mM histamine treatment. Mean ±sem, n = 5–7 assays (75–140 animals) per condition. Mean ±sem. (C) Plasticity of *pgcy-32::HisCl1* day 7 adults treated with 20 mM histamine is significantly higher than *pgcy-*

*Figure 3 continued on next page*

Figure 3 continued

*32::HisCl1* animals without histamine treatment, or control animals with 20 mM histamine treatment. Plasticity indices were normalized to day 1 no histamine treated *pgcy-32::HisCl1* adults. Mean ±sem, **p<0.01, unpaired t-test. (D–E) $O_2$-evoked speed responses of day 4 adults with (D) and without (E) *pgcy-32::*ChR expression treated with all-*trans* retinal (ATR) and blue light from L4 stage to day 3 of adulthood. Mean ±sem, n = 6–7 assays, ##p<0.01 indicates a significant difference of $O_2$-evoked speed responses upon ATR treatment, **p<0.01 next to the data point in the plot indicates a significant difference of speed at this [$O_2$] point, mixed model ANOVA with Holm-Sidak test. (F) Long-term optogenetic stimulation accelerates plasticity decline with age. Mean ±sem, **p<0.01, ns, p>0.05, unpaired t-test.

The online version of this article includes the following figure supplement(s) for figure 3:

**Figure supplement 1.** Inhibition of neuronal activity does not affect the plasticity of day seven adults cultured at 7% $O_2$.

**Figure supplement 2.** Organismal oxidative stress is not responsible for the accelerated plasticity decline with age for 21% $O_2$ cultured animals.

(*Figure 3—figure supplement 1B–F*). NAC significantly reduced expression of the oxidative stress reporter *gst-4p*::GFP, confirming that supplementation with this compound reduces oxidative stress in the assayed strain and culture conditions (*Figure 3—figure supplement 1G*).

Thus, long-term chemogenetic inactivation of $O_2$-sensing neurons prevents the loss of experience-dependent plasticity in ageing animals cultured in high oxygen concentrations, whereas chronic optogenetic activation of these neurons induces a loss of plasticity in ageing animals cultured at low oxygen. Treatment with the antioxidant NAC to reduce oxidative stress does not ameliorate the loss of experience-dependent plasticity in worms cultured at high oxygen. Together, these data indicate that the amount of neuronal activity governs the decline of cognitive plasticity with age: chronically high neuronal activity causes the decline and eventual loss of the ability of neurons to reconfigure their responses following sensory input changes, compromising their ability to generate behavioral plasticity and adaptation in ageing individuals, whereas neurons that are chronically under-active in the long term are better at retaining neuronal plasticity.

## Neuronal activity state dynamically regulates gene expression in the $O_2$-sensing neurons during ageing

Sustained neuronal excitation and the associated elevated intracellular [$Ca^{2+}$] profoundly affect gene transcription, reconfiguring functional properties of neurons and underpinning experience-dependent plasticity (*Flavell and Greenberg, 2008*; *Tyssowski et al., 2018*). To elucidate the molecular and cellular processes responsible for the rapid loss of plasticity with age induced by high neural activity, we investigated whether and how neuronal activity regulates gene transcription in the $O_2$-sensing neurons during ageing. $O_2$-sensing neurons are tonically activated by 21% oxygen, and persistently inactive at 7% $O_2$ (*Busch et al., 2012*). We therefore studied the transcriptomes of neurons from animals cultured in high (21%) or low (7%) $O_2$ concentrations as a proxy for neurons at high and low activity state, respectively. We isolated $O_2$-sensing neurons from 1- and 7-day-old age-synchronized *C. elegans* populations expressing *pgy-37::GFP* in the URX, AQR, and PQR oxygen-sensing neurons by enzymic dissociation of worm bodies and fluorescence-activated cell sorting, and determined their transcriptional profiles by RNA sequencing (*Figure 4A* and Materials and methods; *Supplementary file 1*). For comparison we also obtained transcriptional profiles of whole *C. elegans* bodies at each corresponding condition (*Supplementary file 1*). An interactive presentation of the dataset using the web tool DrEdGE (*Tintori et al., 2020*) is available (*Supplementary file 2*; dredge. bio.unc.edu/BuschLab_eLife). For 1-day-old animals cultured in 21% $O_2$, 7881 genes were differentially expressed in the $O_2$-sensing neuron samples when compared to whole-worm transcriptomes, of which 4058 genes were enriched in the $O_2$-sensing neurons (at a false discovery rate (FDR) of <0.1 and a minimum fold change of 1.5). Genes known to be expressed in URX, AQR, and PQR neurons were among the most highly enriched transcripts in neuronal samples from both 1- and 7-day-old animals. These genes included, for example, the atypical guanylate cyclases *gcy-32, gcy-33, gcy-34, gcy-35, gcy-36* and *gcy-37*, which are causally linked to oxygen sensing (*Cheung et al., 2004*; *Gray et al., 2004*; *Zimmer et al., 2009*), and the FMRFamide neuropeptide gene *flp-8,* which is only expressed in the URX and AUA neurons (*Kim and Li, 2004*). Tissue enrichment analysis (*Angeles-Albores et al., 2016*) of all the neuronal samples also indicated that they were enriched for genes known to be expressed in URX, AQR, and PQR neurons (*Supplementary file 3*). Moreover, a previously published list of genes enriched in the $O_2$-sensing neurons (*Cao et al., 2017*; *Supplementary file 4*) is also significantly enriched in our $O_2$-sensing neuron samples (normalized

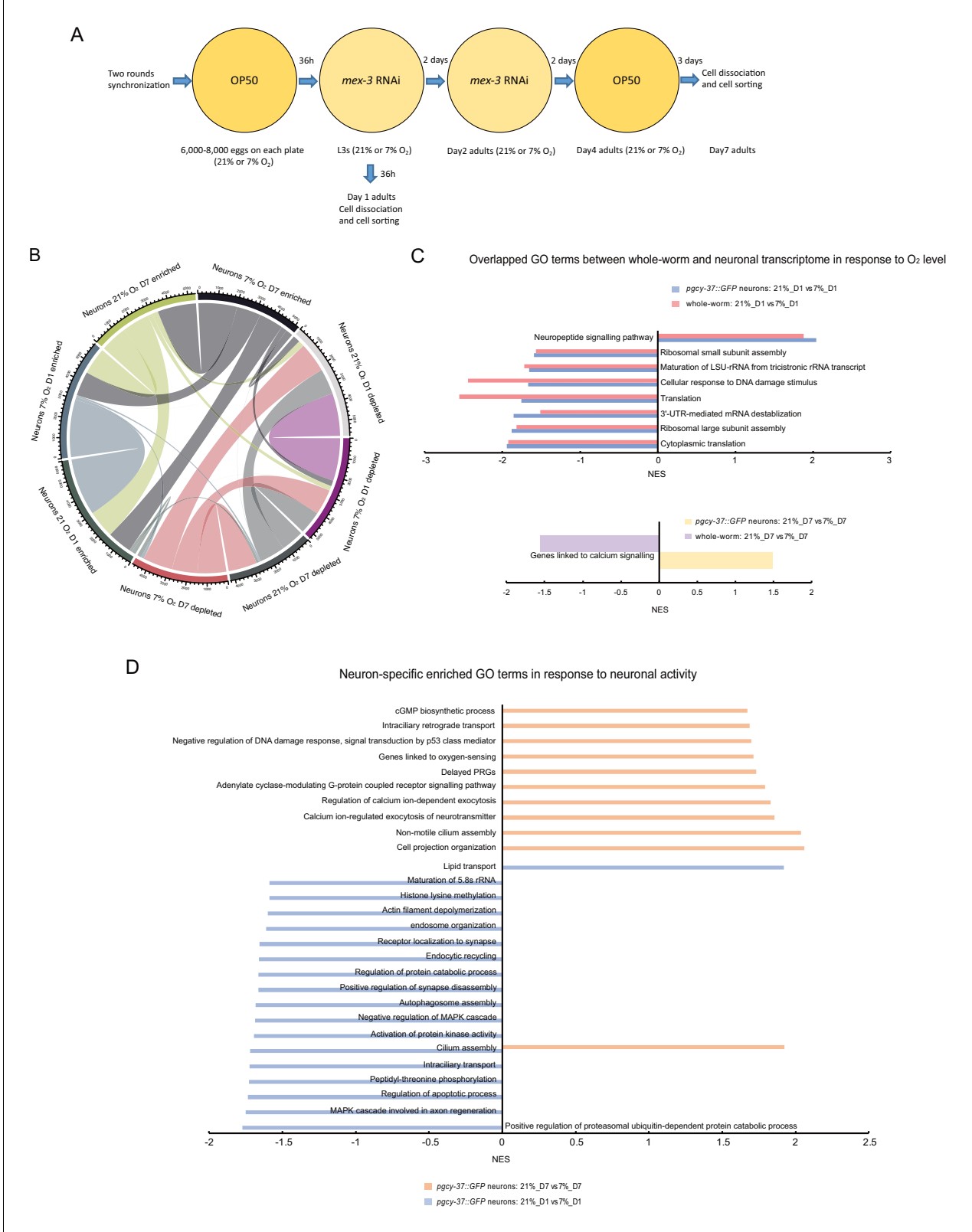

**Figure 4.** Neuronal activity state dynamically alters the expression of specific sets of genes in O$_2$-sensing neurons of 1- and 7-day-old adults. (**A**) Experimental overview of animal preparation and cell sorting for day 1 and day 7 adults. Animals were synchronized by bleaching and the eggs were plated on OP50 plates. L3 stage animals were transferred to *mex-3* RNAi bacterial seeded plates to avoid progeny contamination (see Materials and methods for details). Day four adults were transferred to OP50 plate again until cell dissociation for FACS sorting and RNA preparation.
*Figure 4 continued on next page*

*Figure 4 continued*

(B) Chord diagram showing overlapped enriched and depleted genes across sorted neurons of day 1 and day 7 adults cultured at 21% and 7% $O_2$ respectively. The numbers around the circle signify number of genes. FDR < 0.1 and fold change >1.5. (C) Enriched GO terms (with potential neuronal functions) for $O_2$ level-induced changes that are present in both whole worm and sorted neurons of day 1 and day 7 adults respectively. NES, normalized enrichment score. p<0.05 and FDR < 0.25. (D) Neuron-specific GO terms (with potential neuronal functions) in response to $O_2$-induced neuronal activity in day 1 and day 7 adults. p<0.05 and FDR < 0.25.

The online version of this article includes the following figure supplement(s) for figure 4:

**Figure supplement 1.** Structural and functional validation of the strain for cell-specific RNA-sequencing.

**Figure supplement 2.** Comparison of activity-regulated DEGs between neuronal and whole worm, as well as between day 1 and day 7 samples.

enrichment score = 1.79, q = 0). These data confirmed that neuronal RNA isolated from both the 1- and 7-day-old animals in the experiment was highly enriched for transcripts expressed in the $O_2$-sensing neurons URX, AQR, and PQR. We then did a thorough overlap analysis for differentially expressed genes (DEGs), comparing the $O_2$-sensing neurons to whole-worm-derived transcripts of 1- and 7-day-old animals, and found an extensive overlap of genes enriched or depleted in the $O_2$-sensing neurons across age and $O_2$ culture conditions, suggesting that the expression of genes for neuronal identity remains stable at different culture conditions and life stages (*Figure 4B*).

When comparing high (21%) and low (7%) $O_2$ conditions, 301 genes were found to be differentially expressed among the $O_2$-sensing neuron samples and 248 genes among the whole worm samples in 1-day-old adults, whereas in 7-day-old adults, 264 neuronal and 50 whole-worm genes were differentially expressed. Very few of these DEGs were common to the neuronal and whole-worm gene sets, or to the young and old samples (*Figure 4—figure supplement 2A,B*). To determine which gene categories were enriched in the differentially expressed gene sets, we performed gene ontology (GO) analysis. As expected, several GO terms were co-regulated in the same direction in the neuronal and whole-worm transcriptomes, which may reflect general cellular responses to ambient $O_2$ concentration. Specifically, genes involved in neuropeptide signaling were upregulated in both neurons and whole worm transcriptomes of young adults cultured at the high $O_2$ concentration, whereas several GO terms related to translation were downregulated in both neurons and whole worms at this $O_2$ concentration (*Figure 4C*). In 7-day-old worms, only one gene class overlapped between neurons and whole-worm samples (*Supplementary file 5*). Although the genes expressed in the $O_2$-sensing neurons that are upregulated at high $O_2$ overlap by only nine genes between young and old animals (*Figure 4—figure supplement 2B*), they were enriched in many of the same categories of *C. elegans* phenotypes as shown by phenotype enrichment analysis (*Angeles-Albores et al., 2016*; *Supplementary file 3*). Genes whose expression was significantly increased at 21% $O_2$ were enriched in phenotypes primarily linked to high locomotory activity, such as 'forward locomotion increased' and 'pausing decreased' (*Supplementary file 3*). This is notable as *C. elegans* display persistently high locomotory activity at 21% $O_2$ (*Busch et al., 2012*).

The GO terms that were enriched in genes differentially expressed at high or low $O_2$ concentration specifically in $O_2$-sensing neurons were very different in young and old adult worms: no categories were co-regulated in the same way in both age groups. In neurons from young animals, only lipid transport was upregulated by high $O_2$, whereas most categories that were downregulated were linked to cellular organization and signaling processes (*Figure 4D*). This suggests that the $O_2$-sensing neurons largely do not rely on inducing transcription to maintain high neural activity in young adults. By contrast, in neurons from old animals all enriched gene groups were upregulated at high $O_2$, which signifies chronically high neuronal activity (*Figure 4D*, *Supplementary file 5*). These gene groups were related to $Ca^{2+}$-dependent processes and signaling pathways such as cGMP signaling, which is essential for $O_2$ sensory function (*Couto et al., 2013*; *Figure 4D*). We also found that a manually curated set of 'genes linked to $Ca^{2+}$ signaling' (*Supplementary file 6*), which primarily represents genes that modulate $Ca^{2+}$ homeostasis, was upregulated by high $O_2$ in aged neurons but not in young neurons, and was downregulated in whole worms cultured in high $O_2$ (*Figure 4C*). This suggests that the accumulated effect of chronically elevated $[Ca^{2+}]$ in tonically active $O_2$-sensing neurons increases the expression of $Ca^{2+}$ homeostasis-modulating genes.

In response to elevated neuronal activity, mammalian neurons induce the expression of hundreds of activity-regulated genes (ARGs), which are thought to orchestrate transcription-dependent neuronal plasticity (*Tyssowski et al., 2018*). By looking for orthologs in *C. elegans* of genes in mouse

cultured cortical neurons and the visual cortex (*Tyssowski et al., 2018*), we identified putative ARGs in *C. elegans* $O_2$-sensing neurons (*Supplementary file 6*). Among these ARGs, a subset of delayed primary response genes (dPRGs), whose expression is induced in response to sustained neural activity, was significantly enriched in the set of genes upregulated in high-activity oxygen-sensing neurons from 7-day-old *C. elegans* when compared to low-activity neurons from worms of the same age (*Figure 4D*). Our data suggest that functional ARGs are conserved between *C. elegans* and mammals. Taken together, our findings suggest that neuronal activity regulates gene expression in the $O_2$-sensing neurons during ageing.

## Neuronal activity state alters the ageing trajectory of the neuronal transcriptome in the $O_2$-sensing neurons

We next identified the gene expression changes with age in $O_2$-sensing neurons, and asked whether and how neuronal activity alters the age-related transcriptional change. In both low and high $O_2$ culture conditions, signifying low and high neural activity states, respectively, expression of about 5000 neural genes, ~25% of the genome, significantly changes with age, with about half of them up- or downregulated. In the whole-worm samples, ~2000 genes changed with age, with about half of them up- or downregulated (*Supplementary file 1*). In the $O_2$-sensing neurons, ~60% of the genes differentially expressed with age were shared between the high and low activity states in 21% and 7% $O_2$ cultured animals, respectively (*Figure 5A,D*). There was also a substantial overlap between the DEGs by age in the whole-worm samples in 21% and 7% $O_2$ cultured animals, and between neuronally and whole-worm differentially regulated gene sets (*Figure 5A* and *Figure 5—figure supplement 1A*). This suggests that those overlapping genes constitute a core set of genes changing expression with age. We thus cross-checked the transcriptome changes with age with the 'Meta-Worm' dataset (*Supplementary file 7*) of *C. elegans* ageing signature genes (*Tarkhov et al., 2019*) by gene set enrichment analysis, and found that the subset of MetaWorm genes upregulated with age shows significant enrichment in age-related upregulated genes in both $O_2$-sensing neurons and whole-worm samples; conversely, the subset of MetaWorm genes downregulated with age is significantly enriched in age-related downregulated genes in both neuronal and whole-worm samples (*Figure 5—figure supplement 1B*, *supplementary file 7*). Our data thus likely contain many shared gene expression changes that represent a universal transcriptomic signature of ageing regardless of tissue type and culture condition.

Next, we focused on gene expression changes with age that were specific to the $O_2$-sensing neurons. Consistent with previous findings in the ageing brains of *Drosophila,* killifish, rat, and mouse (*Baumgart et al., 2014*; *Davie et al., 2018*; *Ori et al., 2015*; *Ximerakis et al., 2019*), our data show that translational and ribosome biogenesis related genes are upregulated in aged neurons (*Figure 5C*), comprising the great majority of GO terms upregulated with age. In contrast, translation-related genes in the whole worm transcriptome samples showed consistently decreased expression with age (*Figure 5C*). By pathway analysis (*Supplementary file 8*), eIF2 and eIF4 signaling, required for translation initiation, is upregulated in aged neurons regardless of neuronal activity (*Figure 5E*). Conversely, genes required for neural function are widely downregulated in aged $O_2$-sensing neurons regardless of activity state. GO categories downregulated with age were dominated by neuronal activity-related terms such as adult locomotory behavior, regulation of membrane potential, G-protein-coupled receptor signaling pathway, neuropeptide signaling or chemical synaptic transmission (*Figure 5B*).

To explore why neural plasticity is selectively lost in the ageing $O_2$-sensing neurons experiencing chronically high activity, we then specifically contrasted how the neural transcriptome changes differently with age in animals cultured in 21% or 7% $O_2$ environments, using gene set enrichment and pathway analysis.

We first examined the genes constituting the *C. elegans* $O_2$-sensing machinery by manually curating a gene set based on the literature (*Supplementary file 6*). We found that expression of the $O_2$-sensing machinery in neurons declines with age in line with neural function-related GO terms in general. However, their expression is upregulated with high neuronal activity in day 7 adults' $O_2$-sensing neurons, but not in day 1 adults (*Figure 4D*, *Supplementary file 5*). The aryl hydrocarbon receptor signaling pathway, which controls expression of the soluble guanylate cyclase genes essential for oxygen sensing, is increased with age at high activity, but decreased at low activity (*Figure 5F*). The activity-related genes we identified in *C. elegans* showed a significant decrease in dPRG transcript

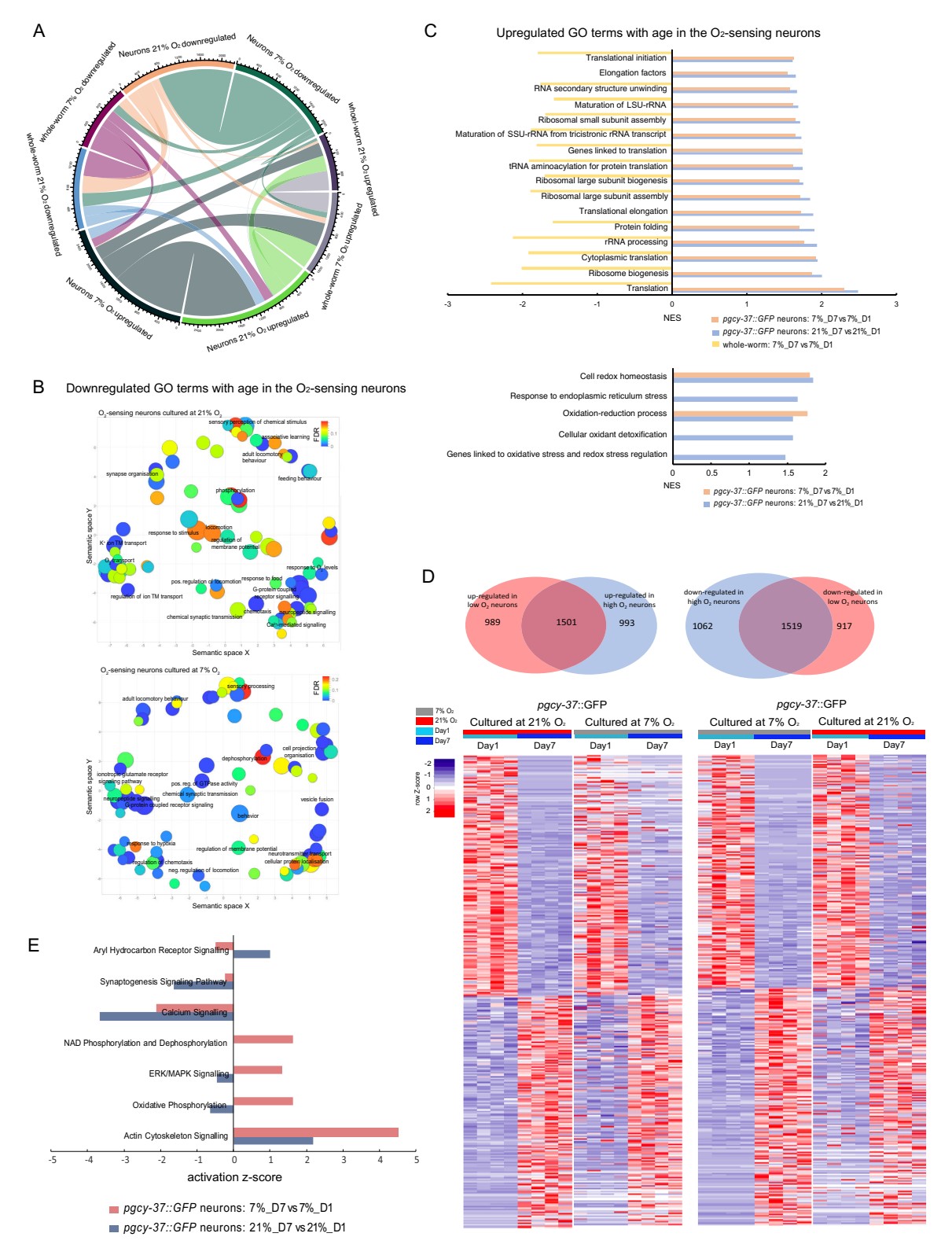

**Figure 5.** Neuronal activity state alters age-related changes in the neuronal transcriptome of O₂-sensing neurons. (**A**) Chord diagram showing overlapped age-related up- and down-regulated genes across whole worm and sorted neurons that cultured at 21% and 7% O₂ respectively. The numbers around the circle signify number of genes. FDR < 0.1 and fold change >1.5. (**B**) GO terms related to the neuronal activity are generally downregulated in aged neurons. REVIGO plot. p<0.05 and FDR < 0.25. (**C**) GO terms related to antioxidant defenses and translation are generally

*Figure 5 continued on next page*

*Figure 5 continued*

upregulated in aged neurons. p<0.05 and FDR < 0.25. (**D**) Venn diagrams and heatmaps showing the comparison of age-related up- and down-regulated genes between low and high activity neurons. FDR < 0.1 and fold change >1.5. (**E**) Canonical pathways with potential neuronal functions with a difference of z-score >1.3 between 21% and 7% $O_2$ exposed sorted neurons from IPA analysis. See *supplementary file 8* for a full list.

The online version of this article includes the following figure supplement(s) for figure 5:

**Figure supplement 1.** Comparison of age-regulated DEGs between neuronal and whole worm samples, and comparison of age-regulated DEGs with MetaWorm gene sets.

abundance in low-activity $O_2$-sensing neurons with age, whereas dPRG expression levels in the high-activity neurons were sustained with age (*Supplementary file 5*). The manually curated set 'genes linked to $Ca^{2+}$ signaling', which primarily represents $Ca^{2+}$ homeostasis-regulating genes, was also specifically down-regulated in low-activity $O_2$-sensing neurons and showed no significant change with age in high-activity neurons (*Supplementary file 5*). Oxidative stress in neurons generally increases with age and causes their greater susceptibility to oxidative insults and disease (*Castelli et al., 2019*). Gene set enrichment analysis shows that oxidative stress-related genes and related GO terms were specifically upregulated with age in the high-activity $O_2$-sensing neurons, whereas only two of these GO terms, such as cell redox homeostasis, showed enrichment with age in the low-activity neurons as well (*Figure 5C* and *Supplementary file 5*); no such increase in any of these GO terms was seen in the whole worm transcriptomes. These results suggest that high-activity neurons devote resources to sustaining tonic neuronal excitation in response to high sensory stimulation, as well as dealing with the resulted high neuronal oxidative stress during ageing.

Conversely, GO terms related to learning and signaling downstream from $Ca^{2+}$ ($Ca^{2+}$-mediated signaling) are significantly downregulated with age only in the high-activity but not the low-activity $O_2$-sensing neurons (*Figure 5B*, *Supplementary file 5*). Pathway analysis shows a stronger downregulation of overall calcium signaling and synaptogenesis signaling pathway with age in the high activity state (*Figure 5E*). These findings suggests that low-activity $O_2$-sensing neurons kept a better 'plasticity reserve' than high-activity neurons during ageing.

Taken together, our data indicate that neural activity alters age-dependent gene expression changes in the $O_2$-sensing neurons. Specifically, chronically high neuronal activity leads to the continued high expression of genes that encode the core $O_2$-sensing machinery, $Ca^{2+}$ homeostasis regulating genes, oxidative stress-related factors, as well as activity-regulated genes, while the expression of learning-related genes and $Ca^{2+}$-mediated signaling factors declined. This suggests that in high activity neurons, resources are directed toward maintaining a high firing rate, leaving limited resources for neural plasticity during ageing.

## Activity-dependent gene expression changes in $O_2$-sensing neurons underpin plasticity regulation during ageing

To examine whether the activity-dependent regulation of transcription in $O_2$-sensing neurons modulates $O_2$-evoked locomotory responses and $O_2$ experience-dependent plasticity, we performed cell-specific RNAi in the $O_2$-sensing neurons to knock down the expression of some of these genes using a transgenic strain that expresses the RNA transporter SID-1 only in these cells, which avoids the RNAi knockdown spreading to other cells or tissues. We first selected 11 genes from top-ranked genes by false discovery rate (FDR) for differential expression in the day 1 adult neuronal transcriptomes, of which four were down- and seven upregulated by neural activity state. Knockdown of 6 of the 11 genes affected either $O_2$-evoked behavior and/or experience-dependent plasticity in 1-day-old adults (*Figure 6—figure supplement 1*). Specifically, knockdown of *cebp-1* (C/EBP transcription factor), *ptp-4* (protein tyrosine phosphatase receptor) or *unc-86* (POU homeodomain transcription factor) in the $O_2$-sensing neurons significantly reduced $O_2$ experience-dependent plasticity (*Figure 6—figure supplement 1G,H,I,M,N*). *nlp-11* neuropeptide, *ptp-4* or *sol-1* (glutamate receptor subunit) knockdown altered both the speed responses to the stepwise changes in $O_2$ concentration of animals kept at 21% $O_2$ and of animals shifted from 21% to 7% $O_2$ (*Figure 6—figure supplement 1A,H,K,N*). Knockdown of *nhr-14* (nuclear hormone receptor) altered speed responses to $O_2$ only in animals kept at 21% $O_2$ throughout (*Figure 6—figure supplement 1D,N*). Together, these results

demonstrate that genes differentially regulated by neural activity in young adults' $O_2$-sensing neurons play a role in the behavior and plasticity evoked by oxygen stimuli.

We then validated the most significant differently expressed genes by neural activity in day 7 neurons. They are *kdin-1*, which encodes the orthologue of the scaffold protein ARMS/kinase D interacting substrate 220 (KIDINS220), *skn-1*, which encodes the *C. elegans* orthologue of the transcription regulator Nrf2, and *cfp-1*, encoding the orthologue of the transcriptional activator CXXC-type zinc finger protein 1. ARMS/KIDINS220 has been implicated in regulating neural excitability and $Ca^{2+}$ homeostasis, and SKN-1/Nrf2 are key regulators of oxidative stress responses and metabolism (*Blackwell et al., 2015*; *Jaudon et al., 2020*). Neural expression of all three genes significantly declines with age in 7% $O_2$ culture but not at 21% $O_2$ (*Supplementary file 1*). We knocked down *kdin-1*, *skn-1* and *cfp-1* by cell-specific RNAi in the $O_2$-sensing neurons only, and tested $O_2$ experience-dependent plasticity in day 2 and day 7 adult worms. *kdin-1* and *skn-1* knockdown did not affect $O_2$-evoked locomotory responses or plasticity in day 2 adults, whereas *cfp-1* RNAi increased $O_2$ responses in day 2 adult animals cultured at high $O_2$, but not in those shifted to 7% $O_2$ for 12 hr, significantly reducing plasticity (*Figure 6A,C*). RNAi of all three genes altered $O_2$-evoked responses in 21% $O_2$-cultured day 7 adults (*Figure 6B*). In particular, RNAi of *kdin-1* restored $O_2$ experience-dependent plasticity in 7-day-old 21% $O_2$ cultured animals, which displayed the same levels of plasticity as at day 2, whereas plasticity significantly declined in control animals (*Figure 6B,C*). RNAi of *skn-1* reduced responses to high $O_2$ in 7-day-old animals kept at 21% $O_2$ throughout, but did not affect the responses of those shifted to 7% $O_2$. *cfp-1* RNAi moderately increased $O_2$ responses of 7-day-old animals shifted to 7% but did not affect those of animals kept at 21% throughout. In contrast, plasticity in 7-day-old animals cultured at 7% $O_2$ was not significantly affected by the knockdown of *kdin-1*, *skn-1* or *cfp-1*, although $O_2$ responses of animals kept at 7% throughout were altered (*Figure 6—figure supplement 2*). These results suggest that genes whose expression is selectively sustained in ageing neurons kept at a high activity state contribute to the loss of $O_2$ experience-dependent plasticity with age. *kdin-1* has a very specific effect on the decline of plasticity without altering $O_2$ responses, whereas *skn-1* and *cfp-1* affect plasticity indirectly through altered $O_2$ responses in non-shifted animals. Thus, our data indicate that activity-induced neuronal transcriptional alteration plays a profound role in experience-dependent plasticity and its decline with age.

## Neuronal $Ca^{2+}$ signaling factors orchestrate the activity-dependent decline of plasticity with age

Our transcriptomic profiling of the $O_2$-sensing neurons suggests that neural activity modulates the changing expression of $Ca^{2+}$ signaling factors with age. Altered $Ca^{2+}$ signaling has been implicated in impaired neural plasticity and age-related cognitive decline (*Foster, 2007*; *Palop et al., 2007*). We therefore hypothesized that $Ca^{2+}$ signaling may be an important intermediary between activity state and the decline of plasticity in neurons with age. To test this and gain an understanding of the molecular processes involved, we characterized how four key $Ca^{2+}$-linked genes, *cmd-1* calmodulin, *tax-6* calcineurin, *unc-68* ryanodine receptor and the *ncx-5* $K^+$-dependent $Na^+/Ca^{2+}$ exchanger are involved in the age-related decline of behavioral plasticity in oxygen responses. All four genes show age-related downregulation in transcription in our data collected from the $O_2$-sensing neurons (*Supplementary file 1*).

The calcium-binding protein calmodulin plays a central role in the regulation of activity-dependent plasticity (*Ma et al., 2014*) and is highly conserved between *C. elegans* and humans with 98% amino acid sequence identity. Due to its pleiotropic roles in many physiological processes, we used cell-specific RNAi to knock down *cmd-1* in URX, AQR and PQR only, and subjected young (day 1) or old (day 7) animals to the environmental shifts detailed in *Figure 1* to test for locomotory speed and experience-dependent plasticity in response to the stepwise changes in oxygen concentrations detailed in *Figure 1—figure supplement 1B*. In 1-day-old adults cultured at 21% $O_2$, *cmd-1* knockdown strongly reduced the locomotory response to 21% $O_2$ after downsteps of $O_2$ concentrations from 21% to 7% $O_2$, but did not diminish the initial locomotory response to high $O_2$ (*Figure 7A*). In animals previously cultured at 7% $O_2$ throughout and also in those shifted from 7% to a 21% $O_2$ environment overnight, *cmd-1* knockdown reduced locomotory responses to high $O_2$ (*Figure 7B*). In contrast, 7-day-old *cmd-1* RNAi adults responded normally to 21% $O_2$ after a period of low $O_2$ exposure (*Figure 7E,F*). These results suggest that CMD-1 is essential in young neurons but not in old neurons to recover from a low activity state to promote a strong avoidance response to high $O_2$.

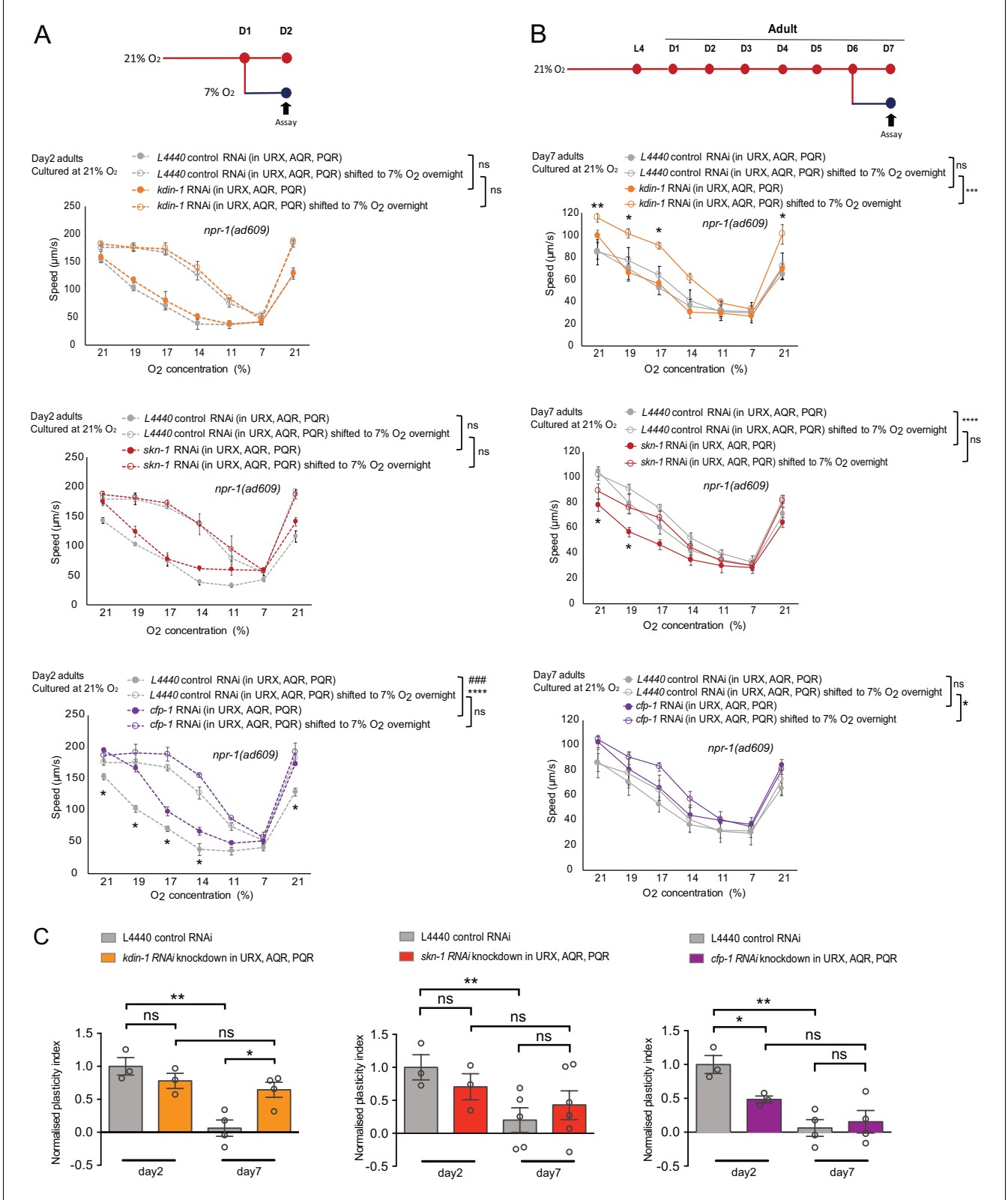

**Figure 6.** Genes selectively downregulated in ageing low-activity neurons underpin plasticity decline with age. (**A–B**) Behavioral test of three DEGs from RNA-seq day 7 neuron samples that specifically downregulated in ageing low-activity neurons. Strain KL92 (*sid-1(pk3321) V.; npr-1(ad609) X.; cipIs7 [pgcy-32::sid-1::SL2-GFP] V*) is used for cell-specific RNAi knockdown by feeding. Mean ±sem, *p<0.05, **p<0.01, ***p<0.001, ****p<0.0001, ns, p>0.05, asterisks above the speed plot indicate a significant interaction between the effect of gene knockdown and [O₂] steps on speed responses, asterisks

*Figure 6 continued on next page*

*Figure 6 continued*

next to the data point in the plot indicate a significant difference of speed at this [$O_2$] point compared to either RNAi feeding or control animals experienced the same $O_2$ culture condition, ###p<0.001 indicates a significant gene knockdown effect on $O_2$-evoked speed responses at that culture condition, mixed model ANOVA with Holm-Sidak test. (C) Normalized plasticity index of *kdin-1*, *skn-1* and *cfp-1* cell-specific RNAi knockdown animals. Mean ±sem, *p<0.05, **p<0.01, ns, p>0.05, unpaired t test.

The online version of this article includes the following figure supplement(s) for figure 6:

**Figure supplement 1.** Functional validation of top gene candidates from RNA-seq day one neuron samples.
**Figure supplement 2.** $O_2$-evoked speed responses and plasticity index of 7% $O_2$ cultured day 7 adults.

Cell-specific knockdown of *cmd-1* in the $O_2$-sensing neurons did not affect experience-dependent plasticity in young adults, but significantly restored experience-dependent plasticity in day 7 adults cultured at 21% $O_2$ (*Figure 7C,G*), where the $O_2$-sensing neurons had been chronically stimulated. Lowering *cmd-1* levels did not further improve the response and plasticity of animals long-term cultured at 7% $O_2$ (*Figure 7F,G*). This suggests that chronically high neuronal activation specifically acts through CMD-1 to result in plasticity loss with age, and that this pathway is not activated when neurons are kept at a low activity state. Lowering neuronal CMD-1 levels can counteract the plasticity decline with age.

To further understand how signaling by $Ca^{2+}$ and calmodulin regulates neural plasticity changes with age, we investigated the $Ca^{2+}$/calmodulin–dependent serine/threonine protein phosphatase calcineurin, the only $Ca^{2+}$-activated protein phosphatase in the brain and a major regulator of synaptic transmission, neuronal excitability and $Ca^{2+}$ homeostasis. Calcineurin is implicated in regulating learning and memory in both mammals and *C. elegans* (*Kuhara and Mori, 2006*; *Mansuy, 2003*). $O_2$-sensing neuron-specific knockdown of *tax-6* in 1-day-old adults altered the response pattern to the stepwise changes in $O_2$ concentration both of animals cultured at 21% $O_2$ throughout and of those shifted to 21% $O_2$ 12 hours before the assay, while animals kept at 7% $O_2$ prior to the experiment behaved normally. Specifically, worms exposed to 21% $O_2$ were less able to slow down at intermediate $O_2$ (between 19% and 14% $O_2$) (*Figure 7A,B*). As a result, *tax-6* RNAi eliminated plasticity in young animals in the 21% to 7% $O_2$ shift assay and significantly reduced plasticity in the 7% to 21% $O_2$ shift experiment (*Figure 7D*).

Like their 1-day-old adult counterparts, 7-day-old adults cultured at 21% $O_2$ with *tax-6* knocked down showed no plasticity, with no significant behavioral difference to controls of the same age (*Figure 7E,H*). Remarkably, 7% $O_2$ cultured animals subject to cell-specific *tax-6* RNAi show significantly less age-related decline of $O_2$-evoked speed responses. Specifically, locomotory speed at high $O_2$ levels increased compared to controls for animals cultured at 7% $O_2$ throughout and in those shifted to 21% $O_2$ 12 hr prior to the assay (*Figure 7F*). Plasticity was retained with age in the same way as in 7-day-old control animals cultured at 7% $O_2$ (*Figure 7H*). The beneficial effect of *tax-6* knockdown was only seen in animals with low neuronal activity in $O_2$-sensing neurons, suggesting that calcineurin signaling contributes to the decline of $O_2$-evoked behavior with age under physiological conditions of stimulation. Moreover, since knockdown of *tax-6* did not restore plasticity in 7-day-old 21% $O_2$ cultured animals, the CMD-1-induced plasticity loss is likely to be acting through a calcineurin-independent pathway. Overall, *tax-6*/calcineurin regulates the tuning of locomotory speed responses to stepwise changes in $O_2$ concentrations and $O_2$ experience-dependent plasticity in young adults, and promotes the decline of locomotory responses with age under physiological conditions.

The ryanodine receptor UNC-68 is an endoplasmic reticulum $Ca^{2+}$ release channel and involved in sustaining tonic sensory responses in the $O_2$-sensing neurons (*Busch et al., 2012*). Cell-specific knockdown of *unc-68* had a similar but weaker phenotype as *tax-6* RNAi in 7-day-old animals, where it increased locomotory responses to $O_2$ in 7% but not 21% $O_2$ cultured animals and likewise did not alter plasticity compared to controls (*Figure 7—figure supplement 1*).

*ncx-5* encodes a member of the $K^+$-dependent $Na^+$/$Ca^{2+}$ exchanger family which plays important roles in sustaining long-term $Ca^{2+}$ homeostasis in many biological processes (*Sharma et al., 2013*). It is orthologous to mammalian NCKX2, which is broadly expressed in the brain and required for long-term potentiation, with NCKX2 null mice showing deficits in learning and memory tasks (*Li et al., 2006*). In line with a previous report (*Sharma et al., 2013*), our $O_2$-sensing neuron-specific

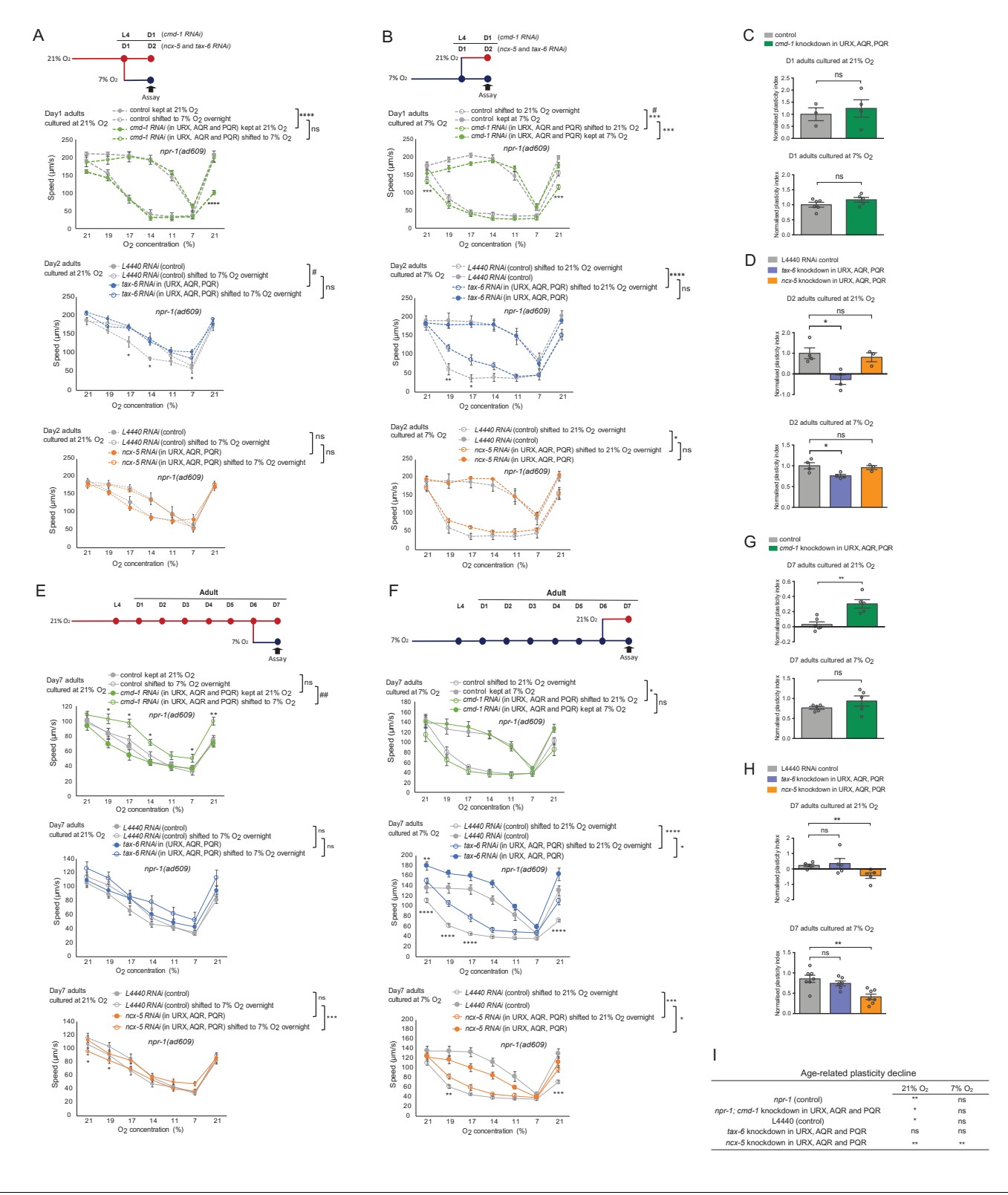

**Figure 7.** Neuronal Ca$^{2+}$ signaling plays an essential role in the activity-dependent decline of plasticity with age. (**A and B**) O$_2$-evoked speed responses of young adults with cell-specific knockdown of *cmd-1*, *ncx-5* and *tax-6* in O$_2$-sensing neurons URX, AQR, and PQR. Strain KL92 (*sid-1(pk3321) V.; npr-1 (ad609) X.; cipIs7[pgcy-32::sid-1::SL2-GFP] V*) is used for cell-specific RNAi knockdown by feeding. Mean ±sem, three or more assays were performed for each condition, *p<0.05, **p<0.01, ***p<0.001, ****p<0.0001, asterisks above the speed plot indicate a significant interaction between the effect of

*Figure 7 continued on next page*

*Figure 7 continued*

gene knockdown and [O₂] steps on speed responses, asterisks next to the data point in the plot indicate a significant difference of speed at this [O₂] point compared to either RNAi feeding or control animals experienced the same O₂ culture condition, #p<0.05 indicates a significant gene knockdown effect on O₂-evoked speed responses at that culture condition, mixed model ANOVA with Holm-Sidak test. (**C and D**) Normalized plasticity index of young adults with cell-specific knockdown of *cmd-1*, *ncx-5* and *tax-6*. Mean ±sem, *p<0.05, ns, p>0.05, unpaired t test. (**E and F**) O₂-evoked speed responses of day 7 adults with cell-specific knockdown of *cmd-1*, *ncx-5* and *tax-6*. Strain KL92 (*sid-1(pk3321) V.; npr-1(ad609) X.; cipIs7[pgcy-32::sid-1:: SL2-GFP] V*) is used for cell-specific RNAi knockdown by feeding. Mean ±sem, five or more assays were performed for each condition, *p<0.05, **p<0.01, ***p<0.001, ****p<0.0001, asterisks above the speed plot indicate a significant interaction between the effect of gene knockdown and [O₂] steps on speed responses, asterisks next to the data point in the plot indicate a significant difference of speed at this [O₂] point compared to either RNAi feeding or control animals experienced the same O₂ culture condition, ##p<0.01 indicates a significant gene knockdown effect on O₂-evoked speed responses at that culture condition, mixed model ANOVA with Holm-Sidak test. (**G and H**) Normalized plasticity index of day 7 adults with cell-specific knockdown of *cmd-1*, *ncx-5* and *tax-6* in O₂-sensing neurons. Mean ±sem, **p<0.01, ns, p>0.05, unpaired t test. (**I**) Table shows the age-related plasticity changes in controls and cell-specific RNAi knockdown animals. Mean ±sem, *p<0.05, **p<0.01, ns, p>0.05, unpaired t test.

The online version of this article includes the following figure supplement(s) for figure 7:

**Figure supplement 1.** O₂-evoked speed responses of animals with *unc-68* cell-specific knockdown in O₂-sensing neurons.
**Figure supplement 2.** O₂-evoked speed responses of day 7 adults with *daf-16* or *crtc-1* specifically knocked down in O₂-sensing neurons.

transcriptomics data show that *ncx-5* is highly enriched in O₂-sensing neurons (**Supplementary file 1**). High-activity neurons have significantly higher *ncx-5* expression than low-activity neurons in day 7 adults, suggesting an involvement in plasticity changes with age. We therefore performed cell-specific RNAi to knock down the expression of *ncx-5* in the O₂-sensing neurons.

In young adults, knockdown of *ncx-5* does not affect the behavior of animals cultured at 21% O₂ (**Figure 7A**), and has a slight but statistically significant effect on the responses to the stepwise changes in O₂ levels of animals shifted from 7% to 21% O₂ overnight, suggesting that *ncx-5* is involved in adaptation to high neuronal activity (**Figure 7B**). The knockdown of *ncx-5* in O₂-sensing neurons did not affect experience-dependent plasticity of young adults, but significantly reduced it in ageing animals (**Figure 7D,H**). Day seven animals shifted from 21% to 7% O₂ for 12 hr significantly reduced rather than increased locomotory responses to O₂, resulting in a reversed plasticity index (**Figure 7E,H**). *ncx-5* RNAi in 7-day-old animals cultured at 7% O₂ significantly decreased behavioral plasticity by reducing oxygen responses in animals kept at 7% and increasing responses in animals shifted to 21% O₂ for 12 hr (**Figure 7F,H**). These results suggest that NCX-5 plays an essential and specific role in sustaining experience-dependent plasticity during ageing and in enabling older animals to adapt to new environments.

We hypothesized that the accelerated behavioral plasticity decline in ageing animals with *ncx-5* knockdown may be a consequence of altered O₂-evoked Ca$^{2+}$ responses in the O₂-sensing neurons of young adults. To test this, we performed Ca$^{2+}$ imaging of the URX O₂-sensing neurons in both 2-day and 7-day-old adults (**Figure 8**, **Figure 8—figure supplement 1A–D**). Cell-specific knockdown of *ncx-5* in the O₂-sensing neurons did not generally alter the response magnitude of the neurons to high O₂ stimuli in either young or old adults (**Figure 8—figure supplement 1C,D**). Intriguingly, although *ncx-5* knockdown did not change the behavioral plasticity of 2-day-old adults, it did reduce plasticity of neuronal Ca$^{2+}$ responses: *ncx-5* RNAi animals shifted from 21% to 7% O₂ for 12 hr increased their Ca$^{2+}$ responses to O₂ stimuli, compared to those cultured at 21% O₂ throughout, less strongly than control animals did upon the same treatment (**Figure 8A,B**, **Figure 8—figure supplement 1A,B**). Also in the converse experiment, control animals shifted from 7% to 21% O₂ for 12 hr very significantly reduced their responses to high and intermediate O₂ stimuli, whereas with *ncx-5* knocked down, URX Ca$^{2+}$ responses to the O₂ stimuli were not significantly different between animals shifted from 7% to 21% O₂ and those cultured at 7% O₂ throughout (**Figure 8C,D**, **Figure 8—figure supplement 1A,B**). Therefore, the ability of the O₂-sensing neurons to adapt to changing O₂ environments requires Ca$^{2+}$ clearance by NCX-5 already in young adults. The accumulated effect of lack of NCX-5 during ageing may be the cause of accelerated behavioral plasticity decline. Furthermore, we found that knockdown of *ncx-5* in URX elevates baseline [Ca$^{2+}$] at 7% O₂ in low O₂ cultured animals, where the neuron is inactive (**Figure 8E**). This suggests that impaired Ca$^{2+}$ homeostasis at rest may be responsible for accelerating the plasticity decline. To further explore this, we tested if knockdown of *cmd-1*, which slows plasticity decline, alters Ca$^{2+}$ homeostasis as well, by performing Ca$^{2+}$ imaging in URX. Ca$^{2+}$ levels were indeed significantly reduced when *cmd-1*

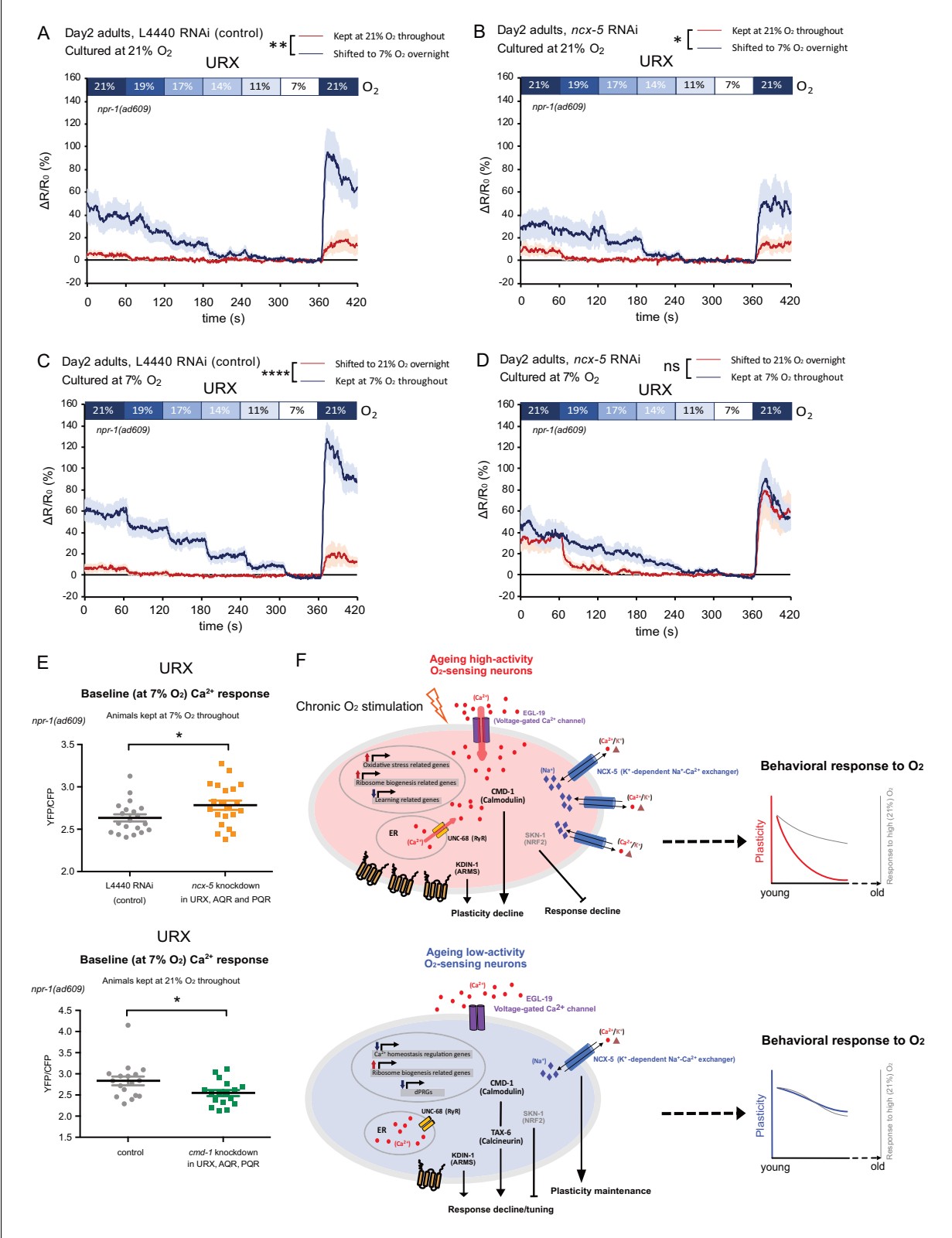

**Figure 8.** NCX-5 and CMD-1 control neuronal Ca$^{2+}$ homeostasis to affect plasticity decline with age. (**A–B**) URX Ca$^{2+}$ responses of day 2 adults cultured at 21% O$_2$ and shifted to 7% O$_2$ overnight with control L4440 (**A**) and *ncx-5* (**B**) RNAi knockdown in O$_2$-sensing neurons. Strain KL317 (*sid-1(pk3321) V; npr-1(ad609) X; cipIs7[pgcy-32::sid-1::SL2-GFP] V; dbEx(pgcy-32::YC3.60 lin-15(+))*) is used for cell-specific RNAi knockdown by feeding followed by Ca$^{2+}$ imaging. Each [O$_2$] step lasts for 1 min. 21–50 s in each 1-min step was used for statistical analysis. Mean ±sem, n = 16–19 animals per condition,

*Figure 8 continued on next page*

Figure 8 continued

*p<0.05, **p<0.01, asterisks indicate a significant effect of overnight $O_2$ level shift on $Ca^{2+}$ responses, mixed model ANOVA. (C–D) URX $Ca^{2+}$ responses of day two adults cultured at 7% $O_2$ and shifted to 21% $O_2$ overnight with control L4440 (C) and *ncx-5* (D) RNAi knockdown in $O_2$-sensing neurons. Each [$O_2$] step lasts for 1 min. 21–50 s in each 1-min step was used for statistical analysis. Mean ±sem, n = 20–21 animals per condition. ****p<0.0001, ns, p>0.05, asterisks indicate a significant effect of overnight $O_2$ level shift on $Ca^{2+}$ responses, mixed model ANOVA. (E) Baseline URX $Ca^{2+}$ response is higher in 7% $O_2$ cultured animals (day two adults) with *ncx-5* knockdown, and lower in 21% $O_2$ cultured animals (day one adults) with *cmd-1* knockdown in $O_2$-sensing neurons. Mean $Ca^{2+}$ response during 21–50 s of 7% $O_2$ step was used for statistical analysis. Mean ±sem, n = 17–21 animals per condition. *p<0.05, unpaired t-test. (F) Models for molecular and cellular processes in ageing high- and low-activity neurons.

The online version of this article includes the following figure supplement(s) for figure 8:

**Figure supplement 1.** URX $Ca^{2+}$ responses to [$O_2$] of animals with *ncx-5* or *cmd-1* specifically knocked down in $O_2$-sensing neurons.

was knocked down in 1-day-old 21% $O_2$-cultured animals (*Figure 8E*, *Figure 8—figure supplement 1E*). Together, these results suggest that the control of $Ca^{2+}$ homeostasis in young adults by NCX-5 in low-activity neurons and CMD-1 in high-activity neurons determines if behavioral plasticity is retained or lost in aged animals.

Previous studies in *C. elegans* have highlighted the role of longevity regulation pathways, in particular of the insulin/insulin-like growth factor signaling(IIS)/FOXO pathway, in altering cognitive performance with age (*Kaletsky et al., 2016*; *Kauffman et al., 2010*; *Murakami et al., 2005*). To test if the IIS/FOXO signaling pathway regulates plasticity decline in the $O_2$-sensing neurons, we performed cell-specific RNAi knockdown of *daf-16* in $O_2$-sensing neurons, but did not observe any significant effect (*Figure 7—figure supplement 2A,B*), suggesting that IIS/FOXO signaling is not involved in the $O_2$ experience-dependent decline of plasticity with age. The TAX-6 calcineurin phosphatase is known to modulate *C. elegans* longevity via the *CREB*-regulated transcriptional coactivator CRTC-1 (*Mair et al., 2011*). However, knockdown of *crtc-1* in the $O_2$-sensing neurons affects neither $O_2$-evoked locomotory responses nor experience-dependent plasticity (*Figure 7—figure supplement 2C,D*). Thus, these data suggest that neuronal activity state affects age-related plasticity decline independent of lifespan-regulating pathways.

Overall, our results indicate that factors linked to neuronal $Ca^{2+}$ signaling govern the decline of the experience-dependent plasticity in $O_2$ responses with age.

## Discussion

As we grow older, the brain functionally deteriorates and is increasingly at risk of neurodegeneration and dementia. Neural plasticity as the basis of learning and memory formation is among the earliest cognitive functions to deteriorate with age, but the underlying molecular and physiological processes are little understood (*Park and Bischof, 2013*). Here, we establish a new behavioral paradigm in *C. elegans* to mechanistically dissect neural plasticity decline with age. Ambient oxygen drives robust and long-lasting neuronal activation and avoidance behavior of *C. elegans* (*Busch et al., 2012*). *C. elegans* also adapts to recently changed oxygen environments on a timescale of hours by showing increased or decreased sensitivity to intermediate oxygen levels, resulting in altered aerotaxis. This reprogramming of the behavior is reflected in plasticity of the URX sensory neurons' response to oxygen stimuli, in which a prior shift from a 21% to a 7% $O_2$ environment strongly sensitizes $O_2$-evoked $Ca^{2+}$ responses in the cell body. The reprogramming of $Ca^{2+}$ responses of the sensory neuron itself suggests a form of intrinsic neuronal plasticity operating in URX, where long-term changes in input cause the persistent modification of its intrinsic electrical properties to adjust their excitability relative to input (*Debanne et al., 2019*). Intrinsic neuronal plasticity is involved in mediating learning and memory and in homeostatic regulation, but few studies have investigated it in vivo (*Mozzachiodi and Byrne, 2010*).

Intriguingly, we found that the decline of $O_2$-evoked plasticity with age is conditional on the $O_2$ environment – in animals cultured at 21% $O_2$, plasticity rapidly declines and is lost within one week of adulthood, whereas at 7% $O_2$ culture, plasticity was sustained even in 10-day-old adults. For comparison, in *C. elegans* chemotaxis, learning and memory abilities start to decline after 2–3 days of adulthood and are lost by the end of the first week (*Kauffman et al., 2010*). The *C. elegans* oxygen-sensing neurons are tonic sensors which show chronically elevated neural activity and cytoplasmic $Ca^{2+}$ for as long as they are exposed to 21% $O_2$ (*Busch et al., 2012*). Our data provide compelling

support for the hypothesis that the behavioral plasticity decline with age is specifically induced by chronic high neuronal activity rather than ambient $O_2$-induced high oxidative stress: long-term chemogenetic inhibition of neuronal activity in the $O_2$-sensing neurons by HisCl1 was sufficient to restore behavioral plasticity in day 7 adults cultured at high oxygen, while chronic optogenetic excitation of the URX $O_2$-sensing neurons strongly impaired behavioral plasticity of day four adults cultured at low oxygen. In contrast, antioxidant treatment did not improve plasticity decline with age, although a reduction of whole worm oxidative stress was observed. Furthermore, in our RNA-seq data, there was little overlap between neuronal and whole-worm differentially expressed genes in response to different $O_2$ levels in both young and old adults (*Figure 4—figure supplement 2A*), which further supports a specific and profound role played by neural activity state in governing plasticity decline with age, distinct from a general effect of $O_2$-induced cellular oxidative stress. Interestingly, 5-day-old worms grown at 21% $O_2$ and switched to 7% $O_2$ for 24 hr showed a substantial recovery of plasticity, which suggests the loss of plasticity due to chronic hyperactivity in the $O_2$-sensing neurons is not permanent and can be restored by prolonged reduction of neural activity even in aged animals.

Our results point to neural excitation and $[Ca^{2+}]$ as key determinants of the decline of neural plasticity with age, where persistently high neural activity state causes neurons to lose their ability to adapt and learn new information, while previously learned responses are maintained. The transcriptomics data suggest that neuronal activity regulates neuronal transcriptome dynamically during ageing. Genes that form the oxygen-sensing machinery and those directly related to oxygen responses such as cGMP signaling; those that defend cells against oxidative stress; genes linked to $Ca^{2+}$ signaling; and activity-regulated genes are all significantly more highly expressed in ageing $O_2$-sensing neurons maintained at chronically high activity, compared to ageing neurons kept at low activity. In contrast, genes required for learning or modulation of synaptic transmission are downregulated with age in the high-activity but not the low-activity neurons. Based on these data, we propose that chronically high activity causes neurons to direct resources toward maintaining a high tonic firing rate, sustaining cellular $Ca^{2+}$ homeostasis, and defending themselves against increased oxidative stress caused by elevated neural activity; leaving fewer resources for the cellular mechanisms required for maintaining neural plasticity. It would be interesting to compare our differential gene expression data with transcriptomics data obtained from regions of ageing human brains associated with high or low neural activity, to determine whether age-related transcriptional changes altered by neuronal activity affect evolutionarily conserved pathways.

Another important finding from the cell-type-specific RNA sequencing is that genes underlying neuronal function, including those involved directly in sensing oxygen, are generally downregulated in the aged $O_2$-sensing neurons, consistent with previous findings in the ageing brains of mouse, *Drosophila* and *Aplysia* (*Davie et al., 2018*; *Greer et al., 2018*; *Ximerakis et al., 2019*). It is not well understood how the downregulation of neuronal activity-related genes relates to ageing processes. A recent study reported that global inhibition of neuronal activity can extend lifespan in humans, mice and *C. elegans* (*Zullo et al., 2019*). Also, lowering neuronal activity reduces tau propagation and β-amyloid deposition (*Wu et al., 2016*; *Yuan and Grutzendler, 2016*). This suggests that the downregulation of neuronal activity does not simply reflect a decline of neuronal function with age but may have a protective role in longevity. Such downregulation would be expected to sacrifice neuronal responsiveness and especially experience-dependent plasticity, which depends on new gene synthesis (*Yap and Greenberg, 2018*). However, the ability of *C. elegans* to generate oxygen-evoked behaviors in 7-day-old adults shows only a modest decline. In that regard, it is noteworthy that the most significant gene classes upregulated in aged neurons are translational and ribosome biogenesis-related genes, in particular the eIF2 and eIF4 signaling pathways which control translational initiation. So far, the purpose of upregulating the neuronal translation machinery with age is unknown. It is possible that it constitutes an adaptation to compensate for the wide-ranging decline of genes required for neuronal function, to sustain continued functioning and sensory responses of the ageing nervous system. This upregulation is neuron-specific, as gene ontology terms related to translation were generally downregulated in the whole-worm samples. Neuron-specific upregulation of translation factors and ribosome biogenesis with age has also been identified in mice (*Ximerakis et al., 2019*), rats (*Ori et al., 2015*), killifish (*Baumgart et al., 2014*), and *Drosophila* (*Davie et al., 2018*), suggesting that it is a highly conserved mechanism in neuronal ageing. This different transcriptional regulation of translational and ribosome biogenesis pathways is consistent with

the idea that nervous system ageing is distinct from that of other tissues. In the future, it will be interesting to investigate whether interfering the age-dependent upregulation of translation and ribosome biogenesis processes affects cognitive performance in ageing animals.

A wealth of studies supports the hypothesis that dysregulation of $Ca^{2+}$ signaling is responsible for the decline of neural and cognitive function with age and increases the risk of neurodegeneration, but the mechanisms involved are little understood (*Berridge, 2011*; *Lerdkrai et al., 2018*; *Pchitskaya et al., 2018*; *Thibault et al., 2007*; *Toescu, 2007*; *Toescu and Verkhratsky, 2007a*; *Toescu and Verkhratsky, 2007b*). Our study determines $Ca^{2+}$ homeostasis control as a critical intermediary between neuronal activity and neuronal ageing, and identifies specific and distinct roles of calmodulin, calcineurin, the NCKX $K^+$-dependent $Na^+/Ca^{2+}$ exchanger and the Kidins220/ARMS scaffold protein in the neuronal activity-dependent plasticity decline with age (*Figure 8F*). All four proteins are involved in regulating neural $Ca^{2+}$ homeostasis (*Bagur and Hajnóczky, 2017*; *Hassan and Lytton, 2020*; *Jaudon et al., 2020*; *Mansuy, 2003*), and NCKX and Kidins220 were not previously known to contribute to cognitive decline. Reduction of calmodulin (*cmd-1*), calcineurin (*tax-6*) and Kidins220 (*kdin-1*) generally had a beneficial effect, promoting plasticity and/or locomotory activity with age, whereas removal of NCKX (*ncx-5*) accelerated the decline of plasticity, suggesting that the decline of plasticity is an actively managed process in which individual $Ca^{2+}$-regulated pathways play different roles.

Our findings suggest a model where in the high neural activity state, intracellular $[Ca^{2+}]$ is chronically elevated, and the resulting overactivation of calmodulin/CMD-1 promotes a gradual decline of plasticity (*Figure 8F*). Knockdown of *cmd-1* restores behavioral plasticity in ageing worms by reducing intracellular $[Ca^{2+}]$ without otherwise changing the $O_2$-evoked speed response, and is likely acting through a calcineurin-independent pathway, as *tax-6* knockdown does not phenocopy the effect of *cmd-1* RNAi. Calcineurin as an early responder to $Ca^{2+}$ is known to act in a negative feedback loop to regulate $Ca^{2+}$ homeostasis, contributing to downregulating $Ca^{2+}$ signaling by weakening $Ca^{2+}$ influx and $Ca^{2+}$ release from intracellular stores (*Mansuy, 2003*). *tax-6* knockdown eliminated plasticity in the highly active $O_2$ neurons of young and old worms, which may be the result of an excessive rise of intracellular $Ca^{2+}$. Likewise, removal of the $Ca^{2+}$ exchanger NCX-5 would impair $Ca^{2+}$ clearance from the cytosol and did indeed not have any beneficial effect on the plasticity loss of the highly active neurons. *ncx-5* expression is upregulated with high activity, presumably in response to the increased demand of extruding intracellular $Ca^{2+}$ from highly active neurons.

In contrast, we found that neuron-specific knockdown of *kdin-1*, whose expression is sustained with age in high-activity neurons but significantly decreases in low-activity neurons, restores experience-dependent plasticity in 7-day-old adults cultured at 21% $O_2$. *kdin-1* is a member of the Kidins220 (kinase D-interacting substrate of 220 kDa)/ankyrin repeat-rich membrane spanning (ARMS) family, which acts as a signaling platform at the plasma membrane. Kidins220 controls neuronal activity and excitability and negatively regulates long-term potentiation in the hippocampus (*Strack et al., 2012*). A recent study in glia cells showed that it regulates intracellular $Ca^{2+}$ homeostasis via store-operated $Ca^{2+}$ entry (*Jaudon et al., 2020*). Tantalizingly, Kidins220 expression is regulated by neural activity levels, and in turn inhibits synaptic plasticity (*Scholz-Starke and Cesca, 2016*; *Wu et al., 2010*). Together, these results suggest an evolutionarily conserved mechanism where persistent expression of Kidins220/*kdin-1* in ageing neurons causes plasticity loss, while plasticity is sustained in those cells that downregulate *kdin-1* expression with age. It will be interesting to explore in future studies if CMD-1 and KDIN-1 act in the same pathway to induce age-related plasticity decline.

In neurons that are at a lower activity state, intracellular $[Ca^{2+}]$ is kept at low levels. Surprisingly, in aged neurons maintained at lower activity, *tax-6* RNAi not only did not decrease plasticity but even improved overall $O_2$-evoked behavioral responses, suggesting, and consistent with previous studies (*Mansuy, 2003*; *Shibasaki et al., 2002*), that calcineurin has multiple distinct roles in regulating excitability and plasticity. Calmodulin/CMD-1 is only partially activated at low activity but may act together with calcineurin/TAX-6 and the ryanodine receptor/UNC-68 to decrease responsiveness in aged neurons. The NCKX/NCX-5 exchanger is responsible for maintaining $Ca^{2+}$ homeostasis by removing $Ca^{2+}$ from the cytoplasm, and we found that its function is required for the plasticity of neurons to adapt to changing $O_2$ levels. Possibly because of the reduced demand for clearing $Ca^{2+}$ from the cytosol compared to high activity neurons, ageing low-activity neurons express less NCX-5.

An important conclusion from our data is that loss of *ncx-5* alters neuronal responsiveness and plasticity of the O$_2$-sensing neurons already in young adults, preceding the defects in behavioral plasticity caused by cell-specific removal of *ncx-5*. The age-related decline in the expression levels of genes mediating neuronal function, including of *ncx-5*, may represent an impairment of a 'cognitive reserve', causing the dissimilar effects of loss of specific neuronal functions on the cognitive performance of young and old individuals. Young adults seem to have higher resilience to a certain level loss of neuronal function than older adults, by compensating for reduced neuronal Ca$^{2+}$ homeostasis for example.

Previous studies have shown that the mechanisms governing the decline of learning and memory in ageing animals intersect with the pathways regulating ageing and longevity (*Kauffman et al., 2010*; *Stein and Murphy, 2012*). It would be interesting in future studies to examine long-lived mutants such as *daf-2* to see whether they are resistant to the decline in plasticity caused by high O$_2$ exposure. Our data on the O$_2$-sensing neuron-specific knockdown of lifespan regulators suggest that neuronal activity state affects age-related plasticity decline independent of lifespan-regulating pathways, in particular of IIS/FOXO signaling. Thus, any effect in long-lived mutants would likely be non-cell-autonomous.

In summary, our study reveals that sustained high neural activity alters the neuronal ageing trajectory and negatively impacts plasticity in the ageing brain. Through Ca$^{2+}$-evoked signaling and transcriptional changes, cellular resources are likely redirected away from sustaining neuronal plasticity and toward maintaining and supporting persistently high excitation. Our study and previous research (*Bell and Hardingham, 2011*; *Berridge, 2011*; *Bezprozvanny, 2010*; *Brini et al., 2014*; *Foster, 2007*; *Pchitskaya et al., 2018*; *Toescu and Verkhratsky, 2007a*) support a model that both too low and too high neuronal activity have negative long-term effects on neuronal heath and plasticity. Treatments that moderately reduce neural activity and correct dysregulated neuronal Ca$^{2+}$ signaling in hyperactive neurons may be beneficial for early-stage intervention in cognitive decline and neurodegenerative conditions.

## Materials and methods

**Key resources table**

| Reagent type (species) or resource | Designation | Source or reference | Identifiers | Additional information |
|---|---|---|---|---|
| Strain, strain background (*E. coli*) | OP50 | CGC | RRID:WB-STRAIN:OP50 | |
| Strain, strain background (*C. elegans*) | AX204 | *Coates and de Bono, 2002* | | *npr-1(ad609) X* |
| Strain, strain background (*C. elegans*) | AX1864 | *Busch et al., 2012* | | *npr-1(ad609) X; dbEx(pgcy-32::YC3.60)* |
| Strain, strain background (*C. elegans*) | KL71 | This study | | *Figure 6*<br>*sid-1(pk3321) him5(e1490)V; npr-1(ad609)X* |
| Strain, strain background (*C. elegans*) | KL123 | This study | | *Figure 4—figure supplement 1*<br>*npr-1(ad609) X; iaIs25[Pgcy-37 ::GFP + unc-119(+)]* |
| Strain, strain background (*C. elegans*) | KL92 | This study | | *Figure 6*<br>*sid-1(pk3321) V.; npr-1(ad609) X.; cipIs7[pgcy-32::sid-1::SL2-GFP] V* |
| Strain, strain background (*C. elegans*) | KL269 | This study | | *Figure 3*<br>*npr-1(ad609)X; cipIs35[pgcy-32:: HisCl1 codon optimized::SL2 GFP]* |
| Strain, strain background (*C. elegans*) | KL217 | This study | | *Figure 7*<br>*npr-1(ad609) X; cipIs31[pgcy-32::cmd-1 sense fragment:: let-868 3'UTR; pgcy-32::cmd-1 antisense fragment::let-868 3'UTR; ccRFP]* |

*Continued on next page*

*Continued*

| Reagent type (species) or resource | Designation | Source or reference | Identifiers | Additional information |
|---|---|---|---|---|
| Strain, strain background (*C. elegans*) | KL317 | This study | | *Figure 8* <br> *sid-1(pk3321) V; npr-1(ad609) X; cipIs7[pgcy-32::sid-1::SL2-GFP] V; dbEx(pgcy-32::YC3.60 lin-15(+))* |
| Strain, strain background (*C. elegans*) | KL325 | This study | | *Figure 3—figure supplement 2* <br> *dvIs19[(pAF15) gst-4p:: GFP::NLS] III; npr-1(ad609) X* |
| Strain, strain background (*C. elegans*) | KL24 | This study | | *Figure 3* <br> *npr-1(ad609) X; Ex (pgcy-32::ChRcodopt-mCitrine)* |
| Commercial assay or kit | PicoPure RNA isolation kit | Arcturus Bioscience | KIT0204 | |
| Commercial assay or kit | Ovation RNA-seq system V2 | Nugen | 7102 | |
| Chemical compound, drug | Histamine | Sigma-Aldrich | H7250 | |
| Chemical compound, drug | All-*trans* retinal | Sigma-Aldrich | R2500 | |
| Chemical compound, drug | N-acetyl cysteine | Sigma | A7250 | |
| Software, algorithm | DinoCapture 2.0 | Dino-Lite Europe | RRID:SCR_019095 | https://www.dino-lite.eu/index.php/en/support/software |
| Software, algorithm | MCQ Gas Blender 100 | MCQ Instruments | | https://www.mcqinst.com |
| Software, algorithm | Matlab | MathWorks | RRID:SCR_001622 | |
| Software, algorithm | Fiji | GitHub | RRID:SCR_002285 | https://fiji.sc/ |
| Software, algorithm | GraphPad Prism | GraphPad | RRID:SCR_002798 | https://www.graphpad.com/scientific-software/prism/ |
| Software, algorithm | Cutadapt | *Martin, 2011* | RRID:SCR_011841 | https://cutadapt.readthedocs.io/en/stable/ |
| Software, algorithm | STAR | *Dobin et al., 2013* | RRID:SCR_015899 | https://github.com/alexdobin/STAR/releases |
| Software, algorithm | FeatureCounts | *Liao et al., 2014* | RRID:SCR_012919 | http://bioinf.wehi.edu.au/featureCounts/ |
| Software, algorithm | DESeq2 | *Love et al., 2014* | RRID:SCR_015687 | |
| Software, algorithm | GSEA | *Subramanian et al., 2005* | RRID:SCR_003199 | https://www.gsea-msigdb.org/gsea/index.jsp |
| Other | 10 μm filter for cell filtration | CellTrics | 04-004-2324 | |

## Strains and *C. elegans* culture

*C. elegans* were grown at 20°C and kept on NGM plates seeded with *E. coli* OP50 bacteria as food source as previously described (*Brenner, 1974*). Previously described strains are AX204 (*npr-1 (ad609) X*) and AX1864 (*npr-1(ad609) X; dbEx(pgcy-32::YC3.60)*) (*Busch et al., 2012*). KL24 (*npr-1 (ad609) X; cipEx41(pgcy-32::ChRcodopt-mCitrine)*) was generated by ballistic gene transfer of a pExpr plasmid containing a Channelrhodopsin-mCitrine fusion gene, both codon optimized for *C. elegans*, under the *gcy-32* promoter for expression in the $O_2$-sensing neurons. KL71 (*sid-1(pk3321) him5(e1490)V; npr-1(ad609)X*) was generated by crossing AX204 with TU3596 (*sid-1(pk3321) him-5 (e1490) V; lin-15B(n744) X*). KL123 (*npr-1(ad609) X; iaIs25[Pgcy-37::GFP + unc-119(+)]*) was generated by crossing AX204 with ZG610 (*iaIs25[Pgcy-37::GFP + unc-119(+)]*). KL92 (*sid-1(pk3321) V.; npr-1(ad609) X.; cipIs7[pgcy-32::sid-1::SL2-GFP] V*) was generated by miniMos injection (*Frøkjær-Jensen et al., 2014*) as follows: A *sid-1* full length genomic fragment was cloned into the miniMos

plasmid pCFJ907, under the *gcy-32* promoter that drives expression in the $O_2$-sensing neurons. The pExpr *sid-1* was injected at 10 ng/μl to AX204, together with co-injection markers pGH8 (*Prab3:: mCherry::unc-54 3'UTR*) at 10 ng/μl, pCFJ90 (*Pmyo-2:mCherry:unc-54 3'UTR*) at 2.5 ng/μl, pCFJ104 (*Pmyo3::mCherry::unc-54 3'UTR*) at 10 ng/μl, pRF4 at 20 ng/μl, and pCFJ601 (*Peft-3::Mos1 transposase*) at 50 ng/μl. 500 μl 25 mg/ml (diluted in water) G418 antibiotic was directly added to the 6 cm NGM plate the day after injection, and the worms were grown at 25°C. Inserted transgenes were identified by screening for surviving animals that lost the co-injection markers as seen by the absence of fluorescent protein expression in neurons and muscle. For identifying the integration sites in the chromosome, inverse PCR was performed as described (*Frøkjær-Jensen et al., 2014*). This strain was then crossed with KL71. KL269 (*npr-1(ad609)X; cipIs35[pgcy-32::HisCl1 codon optimized::SL2 GFP]*) was also generated by miniMos injection, and a codon-optimized single copy of HisCl1 was inserted. KL217 (*npr-1(ad609) X; cipIs31[pgcy-32::cmd-1 sense fragment::let-868 3'UTR; pgcy-32::cmd-1 antisense fragment::let-868 3'UTR; ccRFP]*) was generated by injecting sense and antisense *cmd-1* fragments (40 ng/μl each) cloned from genomic DNA, co-injected with *punc-122:: RFP* at 20 ng/μl followed by UV integration. KL317 (*sid-1(pk3321) V; npr-1(ad609) X; cipIs7[pgcy-32:: sid-1::SL2-GFP] V; dbEx(pgcy-32::YC3.60 lin-15(+))*) was generated by crossing KL92 to AX1864. KL325 (*dvIs19[(pAF15) gst-4p::GFP::NLS] III; npr-1(ad609) X*) was generated by crossing SPC167 (*dvIs19[(pAF15) gst-4p::GFP::NLS] III; skn-1(lax120) IV*) to AX204.

## Behavioral assays

### $O_2$-evoked speed assay

For the day 1 adults assay, L4 larval stage animals were picked to NGM plates seeded with *E. coli* OP50 2 days before, and cultured overnight in a hypoxia chamber for the 7% $O_2$ culture environment, or outside the chamber for the 21% $O_2$ culture environment, both at the temperature of 20°C with around 38% humidity. 20–30 μl of OP50 was seeded on 3.5 cm low peptone assay plates (*Busch et al., 2012*) the day before the assay. On the day of the assay, 15–30 worms were picked to each of the assay plates and assayed 5–10 min after picking. For the worms grown at 7% $O_2$ overnight, the worms were picked to the assay plates within 15 min before the start of the assay. A gas mixture consisting of 0.04% $CO_2$, a variable $O_2$ concentration, and $N_2$ as the balance gas was generated by a gas mixer (MCQ Gas Blender 100) and pumped into a perspex assay chamber through the inlet tube beside the chamber with a flow rate of 120 sccm (standard $cm^3$/min).

The control worms were tested in parallel for each culturing condition on each day of the assay. The behavior of the animals was recorded with a Dino-Lite USB camera/microscope using the Dino-Capture software, and analyzed with a custom-written Matlab script (*Laurent et al., 2015*). Data represent the mean speed for each assay. Error bars represent SEM. Power analysis of the behavioral data showed that the observed effect size is moderate (Cohen's f is around 0.52 based on Pillai's Trace statistic value of 1.278), and to be detected at a power of 0.8, the required sample size was 2.81 per condition. We thus recorded three or more films for each condition independently.

For the day 7 adults assay, L4s were picked to culture plates seeded with a bacterial food lawn 2 days in advance and transferred to fresh plates every 2 days. The night before assay, conditions were shifted according to specific protocols. The plasticity index is defined as $\sum_c \left( \frac{\text{speed of shifted condition} - \text{speed of original condition}}{\text{speed of original condition}} \right)_c$, where c are the different oxygen concentrations assayed. The plasticity index was normalized to young adults (day 1 or day 2) experiencing the same $O_2$ condition shift, that is shift from 21% $O_2$ to 7% $O_2$, or shift from 7% $O_2$ to 21% $O_2$. Adult age is defined relative to L4 larval stage (=day 0).

### Histamine treatment

Experiments were performed as previously described with slight modifications (*Pokala et al., 2014*). 1M histamine stock solution was prepared with $dH_2O$ and added to NGM before pouring to reach a final concentration of 20 mM. Plates were stored in 4°C for no more than one month and were seeded 2 days before use. Parental animals at day 1 – day 2 adult stage were picked to plates containing histamine. L4 stage progeny was transferred to newly seeded histamine plates and then transferred every 2 days. On day 6 of adulthood, animals were transferred to NGM plates without histamine for recovery at either 21% or 7% $O_2$ overnight. Animals were assayed on day 7 of adulthood on plates without histamine. For control, animals without HisCl1 expression were cultured in

the presence of 20 mM histamine, and animals with HisCl1 expression in $O_2$-sensing neurons were grown on NGM plates without histamine. Histamine treatment has no effect on the lifespan of wild-type worms (*Zullo et al., 2019*).

## NAC treatment

100 mM NAC stock solution (diluted in $dH_2O$) was added to NGM before pouring to reach the final concentration of 5 mM. Plates were stored in 4°C and used within one month. Parental animals were grown on OP50 plates. For day 1 adults assay, L4 animals were picked to NAC plates the night before assay. For the day 7 adults assay, L4 animals were picked to NAC plates and transferred to new NAC plates (seeded 2 days before use) every 2 days until assay. The plasticity index was normalized to day 1 adults without NAC treatment.

## Chronic optogenetic stimulation

Parental KL24 or AX204 animals were cultured at 7% $O_2$ and their progeny at L4 larval stage were used for the assay. KL24 worms were preselected for fluorescence in URX neurons only. 30 µl of 5 mM all-*trans* retinal (Sigma) dissolved in ethanol was spread on the bacterial food lawn 1–2 days before use, and 30 µl of ethanol was spread on the control plates without retinal. Animals were irradiated on a custom-built setup with 300 ms pulsed blue light (Luxeon Rebel Blue LEDs, peak wavelength 470 nm) at an intensity of 10 mW/cm$^2$ and a frequency of 0.8 Hz at controlled temperature (20°C). To limit effects of blue light exposure on organismal health, we tested plasticity in 4-day-old adults. Pulsed blue light stimulation started from L4 stage, and on day 3 of adulthood animals were transferred to normal NGM plates without all-*trans* retinal and kept in the dark overnight until assayed on day 4. Animals were transferred to fresh all-*trans* retinal or control plates every day.

## Aerotaxis assay

Aerotaxis assays were done essentially as described previously (*Gray et al., 2004*). For each assay, 150–200 L4s were picked the night before the day 1 adults assay, or then transferred to newly seeded plates every 2 days for the 7-day-old adults assay. For the assay, animals were picked to 9 cm NGM plates seeded two days before assay to form a thin rectangular bacterial food lawn, and a 30 × 15×0.2 mm polydimethylsiloxane (PDMS) microfluidic assay chamber placed over the animals, connected to two syringes at each end to pump 7% $O_2$ or 21% $O_2$, respectively, into the chamber at a pumping rate of 1.5 ml/min. After 30 min, the location of animals in the chamber was recorded with a Point Grey Grasshopper3 CMOS camera mounted on a Leica stereo microscope for distribution analysis and chemotaxis index quantification. The aerotaxis index is defined as (number of animals at high $O_2$ area – number of animals at low $O_2$ area)/total animal number.

## Ca$^{2+}$ imaging

Ratiometric Ca$^{2+}$ imaging was performed on a Nikon Eclipse Ti-E Inverted compound microscope with a CoolLED pE4000 fluorescence light source using a 40x Apochromat λS LWD water immersion lens, following an established protocol (*Busch et al., 2012*).For imaging in day one adults, L4 animals were picked the day before imaging. For day seven adults imaging, L4 animals were picked and transferred to fresh culture plates every 2 days. For imaging, 2% agarose pad (diluted in M9) was used. Individual animals were glued on pad with Dermabond Topical Skin Adhesive, leaving the nose free of glue. A drop of food was put next to the nose, M9 buffer was applied alongside the worm, and glued worms were covered with a polydimethylsiloxane (PDMS) microfluidic chamber. Gases at specific $O_2$ concentrations were generated by a gas mixer (MCQ Gas Blender 100) and collected into individual sealable syringes. 435 nm fluorescence light intensity was set at 1% and exposure time was set at 300 ms. The filter set used was Chroma 59017 for dual CFP/YFP recording. An emission splitter (Cairn Research) was used to separate the cyan and yellow emission light with an FITC/Alexa Fluor 488/Fluo3/Oregon Green dichroic, ET480/40 m filter for CFP emission and ET535/30 m filter for YFP emission. Images were captured with an ORCA-Flash 4.0 V2+ camera at 2 frames per second. Imaging data were analyzed using Neuron Analyzer, a customwritten Matlab program, as previously described (*Laurent et al., 2015*).

## RNAi by feeding

RNAi by feeding was performed as previously described (*Kamath and Ahringer, 2003*) with slight modifications. RNAi plates were supplemented with 100 µg/ml carbenicillin and 1 mM IPTG. Plates were poured freshly and stored at 4℃ for no more than 1 month. Bacterial strains (RNase-deficient *E. coli* HT115) expressing dsRNA were obtained from the Ahringer library and confirmed by sequencing. For the *unc-68* RNAi clone which is not available from the library, an *unc-68* gene fragment cloned from genomic DNA was inserted into L4440 plasmid by double digestion using XhoI and Not I, and this plasmid was then transformed into HT115 bacteria. Primers used for *unc-68* fragment PCR cloning from genomic DNA were CGCGCTCTCGAGGCTACATGGTGATTGCCTCA and TAATTTGCGGCCGCTGCTCCCATTT CACATGAGT. Bacteria were grown on LB plates with carbenicillin and tetracycline at 37℃ overnight. Bacterial liquid culture with 100 µg/ml carbenicillin was grown at 37℃ with shaking at 200 rpm for 6–8 hr. The liquid culture was seeded on the plates, left to dry at room temperature for 2 days. L4 stage animals were picked to the RNAi plate. After 48 hr, adults were transferred to the second RNAi plate seeded with the same bacteria, and their progeny was used for the assays. For RNAi by feeding experiments in young adults we assayed 2-day-old adults instead of day 1 adults because RNAi knockdown is more efficient at this age (data not shown). KL92 was used for behavioral assays and KL317 for Ca$^{2+}$ imaging; in both strains, expression of the RNA transporter *sid-1* was limited to the O$_2$-sensing neurons by transgenic expression from the *gcy-32* promoter to achieve cell-specific knockdown (*Calixto et al., 2010*). All the RNAi feeding animals were cultured at 20℃. The RNAi clones used for RNAi feeding experiments are listed in *Supplementary file 9*.

## Sample preparation for cell-specific RNA-sequencing in day 1 and day 7 adults

Strain KL123 was used for cell-specific RNA-sequencing. URX, AQR and PQR were strongly labeled with GFP in this strain in both young and old adults (*Figure 4—figure supplement 1A,B*). Weaker fluorescent labeling was also occasionally detected in the AVM mechanosensory neuron and the ASI chemosensory neurons (*Figure 4—figure supplement 1A,B* and data not shown). AVM and ASI did not respond to oxygen stimuli (*Figure 4—figure supplement 1C–E*), making it unlikely that they would contribute to activity-dependent transcriptional changes under different oxygen conditions. To avoid contamination of the 7-day adult samples with RNA of their offspring, we fed worms with bacteria inducing RNAi against *mex-3*, which causes embryonic lethality with near 100% penetrance (*Kamath et al., 2001*). We confirmed that the *mex-3* RNAi treatment had no impact on either experience-dependent plasticity or the differential decline of plasticity in 7-day-old animals (*Figure 4—figure supplement 1F–H*).

Animals were synchronized by bleaching for two rounds. Eggs were plated on OP50 plates and L3 stage animals were transferred to *mex-3* RNAi plates to avoid egg hatching. To obtain day 7 adults, adults at day 2 were washed off by M9 and transferred to new *mex-3* RNAi plates for further cultivation, and at day 4 were washed off and transferred to new OP50 plates (*Figure 4A*).

Cell dissociation and FACS sorting procedures were performed as previously described (*Kaletsky et al., 2016*) with modifications. For each sample, 10,000–15,000 worms were washed off by M9 and transferred to 1.5 ml Eppendorf tubes, followed by washing with M9 buffer five times. 300 µl lysis buffer (200 mM DTT, 0.25% SDS, 20 mM HEPES pH 8.0, 3% sucrose) was added and worms were incubated for 5–6 min until the head of worms became blunt. Worms were then washed rapidly with M9 buffer five times. 200 µl room temperature pronase (15 mg/ml, dissolved in 340mOsm L15) was added and worms were incubated for 8–15 min. During pronase incubation, a pellet pestle motor was used every 2–3 min for 10 s each time to dissociate and homogenize worm cells. After each round of using the pellet pestle motor, dissociation was monitored by checking 2 µl of the pronase suspension under the microscope. When the majority of big worm chunks were no longer visible, 800 µl ice-cold BSA (0.2%)/L-15 (340mOsm) was added to stop the reaction. At this point, 10 µl of the dissociated worm solution of each sample was transferred to 300 µl RNA extraction buffer (PicoPure RNA isolation kit (Arcturus Bioscience)) and kept on ice. This solution was subsequently used to determine the whole-worm transcriptome. Cells were pelleted by centrifugation at 9600 g for 5 min at 4℃ and then resuspended using 1 ml ice-cold BSA (0.2%)/L-15 (340mOsm), and immediately passed through a 10 µm filter for sorting.

For FACS sorting, age-matched AX204 worms were used as negative control for setting gates. For KL123 (*npr-1(ad609) X; iaIs25[Pgcy-37::GFP + unc-119(+)]*), 5'000–20'000 GFP positive cells from each sample were sorted into 300 µl extraction buffer for RNA extraction using the PicoPure RNA isolation kit (Arcturus Bioscience). RNA samples with a RIN score >7.0 were used for cDNA synthesis. cDNA was generated using the Ovation RNA-seq system V2. Libraries were generated using TruSeq DNA Nano gel free library preparation kit. NovaSeq 50PE was used for all samples. Four independent biological replicates per condition were sequenced.

### RNA-seq data analysis

Reads were trimmed using Cutadapt (version cutadapt-1.9.dev2) (*Martin, 2011*). Reads were trimmed for quality at the 3' end using a quality threshold of 30 and for adapter sequences of the TruSeq DNA kit (AGATCGGAAGAGC). The reference used for mapping was the *Caenorhabditis elegans* (build WBcel235) genome from Ensembl. Reads were aligned to the reference genome using STAR (version 2.5.2b) (*Dobin et al., 2013*) specifying paired-end reads and the option `--outSAM-type` BAM Unsorted. Reads were assigned to features of type 'exon' in the input annotation grouped by gene_id in the reference genome using featureCounts (version 1.5.1) (*Liao et al., 2014*). Genes with biotype rRNA were removed prior to counting. featureCounts assigns counts on a 'fragment' basis as opposed to individual reads such that a fragment is counted where one or both of its reads are aligned and associated with the specified features. Strandedness was set to 'unstranded' and a minimum alignment quality of 10 was specified. The raw counts were filtered to remove genes consisting predominantly of near-zero counts, filtering on counts per million (CPM) to avoid artefacts due to library depth. Differential gene expression analysis was performed using DESeq2 (*Love et al., 2014*) (version 1.18.1 with R version 3.4.2) at a default false discovery rate (FDR) of 0.1 as used in previous studies (*Thomson et al., 2017*), and additionally with a minimal fold change of 1.5 to exclude smaller changes in transcript abundance mostly seen in the whole-worm samples. Pathway analysis was performed using Ingenuity Pathway Analysis (Qiagen). Gene set enrichment analysis was performed in GSEA (Broad Institute) and WormBase Tissue Enrichment Analysis (*Angeles-Albores et al., 2016*). GSEA version 2–2.2.1 was used and gene sets containing more than 500 genes or less than five genes were excluded. Only gene sets with p<0.05 and q < 0.25 were considered as significantly enriched, as done in previous studies (*Ximerakis et al., 2019*). The normalized enrichment score (NES) was defined previously (*Subramanian et al., 2005*).

### Quantitative fluorescence microscopy

KL325 was used for *gst-4p::GFP* fluorescence intensity quantification. NAC (5 mM) treatment was initiated from hatching. Worms were immobilized by 100 mM $NaN_3$ on a 2% agarose pad. Images were captured on a Zeiss Axio Imager Z1m with a 10x objective, and images analyzed by ImageJ. Specifically, a threshold was applied to each picture to highlight the fluorescent whole worm and the mean value of this highlighted area was used for statistical analysis.

### Statistical analysis

Unpaired t-test with Holm-Sidak correction for multiple comparisons was performed to examine the plasticity difference between 21% and 7% $O_2$ animals at day 4, day 7, and day 10 of adulthood. For $O_2$-evoked speed responses, mixed model ANOVA was performed to examine the effect of condition shift (21% to 7% $O_2$ or 7% to 21% $O_2$ the night before assay) and [$O_2$] steps (21%, 19%, 17%, 14%, 11%, 7%, and 21% $O_2$) on worms' speed responses. Mann-Whitney U test was used for aerotaxis index comparison. For $O_2$-evoked speed responses in RNAi feeding assays, mixed model ANOVA with Holm-Sidak test was conducted to examine the effect of gene knockdown and [$O_2$] steps on speed responses. $O_2$-evoked $Ca^{2+}$ responses were analyzed using mixed model ANOVA, and the time period used for analysis is indicated in corresponding figure legends. Plasticity indexes were compared by unpaired t-test. One-way ANOVA with Holm-Sidak test was used for fluorescence intensity comparison. GraphPad Prism was used for statistical analysis.

## Acknowledgements

We thank Sophie Thomson, Sang Seo and Emily Osterweil for helpful advice on RNA-seq analysis; Edinburgh Genomics for RNA-seq library preparation and sequencing; Martin Waterfall for FACS;

and Maria Doitsidou, Sebastian Greiss and Matt Nolan for helpful suggestions. Some strains were provided by the Caenorhabditis Genetics Center, which is funded by NIH Office of Research Infrastructure Programs (P40 OD010440). QL was supported by a University of Edinburgh Global Research Scholarship and a Principal's Career Development Scholarship. We gratefully acknowledge financial support by the Wellcome Trust (109614/Z/15/Z) and the Medical Research Council (MR/N004574/1) to KEB; and the UK Dementia Research Institute, which receives its funding from DRI Ltd, funded by the UK Medical Research Council, Alzheimer's Society, and Alzheimer's Research UK, and the European Research Council (ERC) under the European Union's Horizon 2020 research and innovation programme under grant agreement No 681181 to TS-J.

## Additional information

### Competing interests

Tara L Spires-Jones: Tara Spires-Jones receives funding from 3 industrial collaborators and is on the Scientific Advisory Board of Cognition Therapeutics. None of these influenced the current paper. The other authors declare that no competing interests exist.

### Funding

| Funder | Grant reference number | Author |
| --- | --- | --- |
| Medical Research Council | MR/N004574/1 | Karl Emanuel Busch |
| Wellcome Trust | 109614/Z/15/Z | Karl Emanuel Busch |
| UK Dementia Research Institute | | Tara L Spires-Jones |
| EU Framework Programme for Research and Innovation H2020 | 681181 | Tara L Spires-Jones |

The funders had no role in study design, data collection and interpretation, or the decision to submit the work for publication.

### Author contributions

Qiaochu Li, Conceptualization, Data curation, Formal analysis, Validation, Investigation, Visualization, Methodology, Writing - original draft, Writing - review and editing; Daniel-Cosmin Marcu, Formal analysis, Methodology, Writing - review and editing; Ottavia Palazzo, Investigation; Frances Turner, Declan King, Tara L Spires-Jones, Formal analysis, Writing - review and editing; Melanie I Stefan, Formal analysis, Funding acquisition, Writing - review and editing; Karl Emanuel Busch, Conceptualization, Formal analysis, Supervision, Funding acquisition, Investigation, Visualization, Methodology, Writing - original draft, Writing - review and editing

### Author ORCIDs

Qiaochu Li https://orcid.org/0000-0001-8032-9110
Daniel-Cosmin Marcu https://orcid.org/0000-0002-1006-3094
Tara L Spires-Jones https://orcid.org/0000-0003-2530-0598
Melanie I Stefan https://orcid.org/0000-0002-6086-7357
Karl Emanuel Busch https://orcid.org/0000-0001-7886-3226

### Decision letter and Author response

Decision letter https://doi.org/10.7554/eLife.59711.sa1
Author response https://doi.org/10.7554/eLife.59711.sa2

## Additional files

### Supplementary files

• Supplementary file 1. RNA-seq data for whole-worm and $O_2$-sensing neurons isolated from day 1 and day 7 adults cultured at 21% and 7% $O_2$. The comma-separated file shows gene-wise differential expression log fold changes and false discovery rates.

• Supplementary file 2. DrEdGe files for an interactive presentation of the dataset in *Supplementary file 1*.

• Supplementary file 3. Tissue and phenotype enrichment analysis.

• Supplementary file 4. Previously reported genes enriched in the $O_2$-sensing neurons.

• Supplementary file 5. Significant GO terms from gene set enrichment analysis.

• Supplementary file 6. Manually curated gene sets for GSEA analysis.

• Supplementary file 7. MetaWorm data set of genes differentially regulated with age, with gene set enrichment analysis.

• Supplementary file 8. Activation z scores of pathways identified in the ingenuity pathway analysis.

• Supplementary file 9. List of RNAi clones used in this study.

• Transparent reporting form

### Data availability

Sequencing data have been deposited in GEO under the accession code GSE152680.

The following dataset was generated:

| Author(s) | Year | Dataset title | Dataset URL | Database and Identifier |
|---|---|---|---|---|
| Li Q, Turner F, Busch KE | 2020 | RNA-seq profiling for *C. elegans* O2-sensing neurons at different age and activity states | https://www.ncbi.nlm.nih.gov/geo/query/acc.cgi?acc=GSE152680 | NCBI Gene Expression Omnibus, GSE152680 |

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
