## [Decision Letter]

**Acceptance summary:**

As humans and other animals age, their capacity for neuronal plasticity-a process critical to learning and memory-declines. Using the nematode *C. elegans*, the authors show here that sustained hyperactivity of a specific neuron type accelerates this decline. Together with other work, this suggests that healthy aging relies on a 'sweet spot' of neural activity; levels that are too high may divert resources away from programs that maintain plasticity as neurons age.

**Decision letter after peer review:**

Thank you for submitting your article "High neural activity accelerates the decline of cognitive plasticity with age in *C. elegans*" for consideration by *eLife*. Your article has been reviewed by three peer reviewers, and the evaluation has been overseen by a Reviewing Editor and Piali Sengupta as the Senior Editor. The following individual involved in review of your submission has agreed to reveal their identity: Thorsten Hoppe (Reviewer #1).

The reviewers have discussed the reviews with one another and the Reviewing Editor has drafted this decision to help you prepare a revised submission.

Summary:

In your paper, you examine the effects of persistent neuronal activity, using the *C. elegans* oxygen-sensing neurons as a model, on neuronal plasticity and behavior. Interestingly, chronic exposure to high oxygen, but not low oxygen, leads to age-related declines in neural plasticity. Your paper also provides evidence that this is a specific consequence of chronic activity of the O2 neurons, but we would like this idea to be further tested (see below). Further, using cell-specific transcriptomics and knockdown experiments, you identified specific molecular mechanisms that contribute to the enhancement of age-related decline. The reviewers found these to be very interesting findings that provide new insight into the molecular mechanisms underlying neuronal plasticity in aging.

Essential revisions:

Before your work is suitable for publication in *eLife*, there are several points that must be addressed.

1) There is concern about whether the effects you see on gene expression and behavior are truly due to increased activity of the oxygen-sensing neurons, rather than other effects of high oxygen exposure. For example, some of these could result from general cellular adaptations to high oxygen; further, stimulation of the oxygen-sensing neurons leads to increased cytoplasmic cGMP, which could itself have a significant impact on gene expression. Your revised manuscript should directly test whether increased neuronal activity itself is the cause of the changes you observe after chronic high-oxygen exposure. The reviewers feel that the best way to do this would be to examine the effects of optogenetic stimulation of oxygen-sensing neurons in a 7% oxygen environment.

2) Some interpretations of the calcium-imaging experiments are unconvincing, owing to high levels of noise and small sample size. Because of this, it is not clear that the decline of behavioral plasticity that you report is also occurring at a physiological level. Specifically, in Figure 2C and D, you claim that while there is no plasticity in panel C, there is instead in panel D. However, the noise is so large in these recordings that this interpretation becomes questionable. In Figure 2F, the mean of YA at D1 kept at 7% O_2_ (gray triangles) seems higher due to three data points that appear to be outliers. Therefore, the difference between 21% and 7% groups could be much smaller and may even be not significant. The same applies to Figure 2E. In your revised manuscript, these concerns should be addressed by carrying out additional calcium imaging experiments.

3) In Figure 3, it is not clear whether the behavior of worms with silenced O_2_ sensing neurons was assayed at 7% O_2_. This control seems to be missing from the manuscript. Please add it or explain why it is not necessary.

4) The genetic background used for RNAi knock-down is not clearly specified in the text or the figures. These experiments need to be done in a way that avoids RNAi spreading (e.g., in a *sid-1* mutant background). Please clarify this.

---

## [Author Response]

Essential revisions:1) There is concern about whether the effects you see on gene expression and behavior are truly due to increased activity of the oxygen-sensing neurons, rather than other effects of high oxygen exposure. For example, some of these could result from general cellular adaptations to high oxygen; further, stimulation of the oxygen-sensing neurons leads to increased cytoplasmic cGMP, which could itself have a significant impact on gene expression. Your revised manuscript should directly test whether increased neuronal activity itself is the cause of the changes you observe after chronic high-oxygen exposure. The reviewers feel that the best way to do this would be to examine the effects of optogenetic stimulation of oxygen-sensing neurons in a 7% oxygen environment.

We thank you for this conducive suggestion. In Figure 3 of the first submission, we showed, using chemogenetic silencing of the O_2_-sensing neurons, that high neural activity is *necessary* to accelerate the decline of plasticity. However, we did not demonstrate that high neural activity is *sufficient* to accelerate cognitive decline. We have now performed chronic optogenetic stimulation of the URX O_2_-sensing neurons in a 7% O_2_ environment, using Channelrhodopsin, and find that this treatment significantly reduces behavioural plasticity during ageing, while control animals show no significant decline. The chronic stimulation only alters the behaviour of animals that are switched to 21% O_2_ but not that of animals kept at 7% O_2_ throughout, which suggests that chronic activity of URX does not alter O_2_ responses per se, but specifically changes how the plasticity of the O_2_ response declines with age. These new data have been added to Figure 3.

2) Some interpretations of the calcium-imaging experiments are unconvincing, owing to high levels of noise and small sample size. Because of this, it is not clear that the decline of behavioral plasticity that you report is also occurring at a physiological level. Specifically, in Figure 2C and D, you claim that while there is no plasticity in panel C, there is instead in panel D. However, the noise is so large in these recordings that this interpretation becomes questionable. In Figure 2F, the mean of YA at D1 kept at 7% O_2_ (gray triangles) seems higher due to three data points that appear to be outliers. Therefore, the difference between 21% and 7% groups could be much smaller and may even be not significant. The same applies to Figure 2E. In your revised manuscript, these concerns should be addressed by carrying out additional calcium imaging experiments.

We thank you for this raising this point. We performed power calculation on the previously submitted Ca^2+^ imaging data for the comparison group showing the smallest effect (D7LT7ON21 vs. D7LT7ON7 at 14% [O_2_], Figure 2D) and the observed power is 0.56 at the detected effect size (d=0.81). For a power of 0.8, 21 recordings per group would be required to claim significance at that effect size. We therefore have done further Ca^2+^ imaging recordings, incorporated in Figure 2 of the resubmission, reaching at least 25 recordings per condition. These data confirm the previous finding that there is no plasticity of Ca^2+^ responses in 7-day-old URX neurons when cultured at 21% O_2_, but that those cultured at 7% O_2_ retain plasticity. In the combined results, *p* values decrease for all three conditions that did show plasticity in the URX Ca^2+^ response (D1 adults kept at 21% O_2_ (panel A), D1 adults kept at 7% O_2_ (panel B) and D7 adults kept at 7% O_2_ (panel D)). In contrast, the *p* value for D7 adults kept at 21% O_2_ (panel C) still shows no significant difference:

**Author response table 1. resptable1:** 

	Panel A	Panel B	Panel C	Panel D
*p* values of recordings in first submission:	0.0025 (**)	0.0459 (*)	0.7609 (ns)	0.0478 (*)
*p* values with new recordings included:	0.0004 (***)	0.006 (**)	0.6357 (ns)	0.0031 (**)

The additional recordings therefore significantly strengthen our conclusion that plasticity of URX responses is lost with age only in animals cultured at 21% O_2_, but not in those cultured at 7% O_2_.

Concerning the outliers (that is, neurons responding particularly strongly to the stimulus) in the Ca^2+^ responses of 1-day-old adults, removing them from both the 21% and 7% groups does not change the outcomes of the statistical comparison. Specifically, the scatterplots in Author response image 1 show the combined old and new data with day 1 outliers removed (cf. Figure 2E, F in the first submission, and Figure 2—figure supplement 1C, D in the resubmission). Removing the outliers decreases the *p* value of the comparison between the first and third column of the left panel in Author response image 1 (cf. Figure 2—figure supplement 1D in the resubmission), which demonstrates that the significantly higher mean of D1 young adults kept at 7% O_2_ (gray triangles) compared to that of D1 adults kept at 21% O_2_ (grey dots) is not due to an outlier effect. In the panel on the right (cf. Figure 2—figure supplement 1C in the resubmission), removing outliers does not alter the outcomes of the statistical comparisons.

3) In Figure 3, it is not clear whether the behavior of worms with silenced O_2_ sensing neurons was assayed at 7% O_2_. This control seems to be missing from the manuscript. Please add it or explain why it is not necessary.

We have now carried out the same silencing experiment shown in Figure 3 also in animals cultured at 7% O_2_. The results are shown in Figure 3—figure supplement 1A-C. We find that silencing the O_2_-sensing neurons at 7% O_2_ does not alter the retention of plasticity with age. This finding further corroborates our argument that the O_2_-sensing neurons retain plasticity with age if they are at chronically low activity or silenced.

4) The genetic background used for RNAi knock-down is not clearly specified in the text or the figures. These experiments need to be done in a way that avoids RNAi spreading (e.g., in a sid-1 mutant background). Please clarify this.

We have indeed performed all RNAi knock-down experiments by bacterial feeding in a *sid-1* mutant background to avoid systemic spreading of dsRNA. We have now highlighted this information in both the main text (Results) and the legends of Figures 6-8.